# Unleashing the Power of Data Tsunami: A Comprehensive Survey on Data Assessment and Selection for Instruction Tuning of Language Models

**Yulei Qin**                                                                    *yuleiqin@tencent.com*
*Tencent YouTu Lab*

**Yuncheng Yang**                                                         *yaphabates@sjtu.edu.cn*
*Tencent YouTu Lab, Shanghai Jiao Tong University*

**Pengcheng Guo**                                               *guopengcheng1220@gmail.com*
*Tencent YouTu Lab*

**Gang Li**                                                                         *gang.li@njust.edu.cn*
*Tencent YouTu Lab*

**Hang Shao**                                                                 *talkkingsh@gmail.com*
*Tencent YouTu Lab*

**Yuchen Shi**                                                               *oldcitystal@gmail.com*
*Tencent YouTu Lab*

**Zihan Xu**                                                                       *ianxxu@tencent.com*
*Tencent YouTu Lab*

**Yun Gu**                                                                              *yungu@ieee.org*
*Shanghai Jiao Tong University*

**Ke Li**                                                                         *tristanli.sh@gmail.com*
*Tencent YouTu Lab*

**Xing Sun**                                                                 *winfred.sun@gmail.com*
*Tencent YouTu Lab*

**Reviewed on OpenReview:** `https://openreview.net/forum?id=RJT1baPhdV`

## Abstract

Instruction tuning plays a critical role in aligning large language models (LLMs) with human preference. Despite the vast amount of open instruction datasets, naively training a LLM on all existing instructions may not be optimal and practical. To pinpoint the most beneficial datapoints, data assessment and selection methods have been proposed in the fields of natural language processing (NLP) and deep learning. However, under the context of instruction tuning, there still exists a gap in knowledge on what kind of data evaluation metrics can be employed and how they can be integrated into the selection mechanism. To bridge this gap, we present a comprehensive review on existing literature of data assessment and selection especially for instruction tuning of LLMs. We systematically categorize all applicable methods into quality-based, diversity-based, and importance-based ones where a unified, fine-grained taxonomy is structured. For each category, representative methods are elaborated to describe the landscape of relevant research. In addition, comparison between the  latest methods is conducted on their officially reported results to provide

in-depth discussions on their limitations. Finally, we summarize the open challenges and propose the promosing avenues for future studies. All related contents are available at `https://github.com/yuleiqin/fantastic-data-engineering`.

# 1 Introduction

One of the ultimate goal of developing large lnguage models (LLMs) is to unlock their potentials of generalization to unseen natural language processing (NLP) tasks. Towards this goal, a series of LLMs such as GPTs [Brown et al. (2020); Achiam et al. (2023)], LLaMAs [Touvron et al. (2023a;b); AI@Meta (2024)], and Mistrals [Jiang et al. (2023a; 2024a)] have delivered high-level text understanding and generation capabilities via utilizing vast amount of high-quality web and human-annotated datasets for pre-training and preference alignment [Liu et al. (2023a; 2024c); Sun et al. (2024b); Edunov et al. (2019); Dong et al. (2019)]. During preference alignment, instruction tuning plays an important role in refining the pre-trained LLMs to provide accurate, pertinent, and harmless responses on a collection of downstream tasks [Wei et al. (2021); Sanh et al. (2021); Zhang et al. (2023d); Peng et al. (2023); Longpre et al. (2023); Shu et al. (2023); Jang et al. (2023); Ghosh et al. (2024); Kung & Peng (2023)]. For efficient and effective instruction tuning, existing studies [Ouyang et al. (2022); Taori et al. (2023); Zhou et al. (2024a); Xia et al. (2024a)] have noticed that improving the quality of instruction tuning data (e.g., formulation of well-defined and complete contexts), rather than simply piling up instructions without analysis (e.g., exhaustive collection of open datasets), is of prioritized concerns.

In this work, we aim to unify a wide array of data assessment and selection methods under the context of instruction tuning of LLMs. As revealed from the probabilistic view [John & Draper (1975); Murphy (2012); Albalak et al. (2024)], the statistical patterns inherent in datasets determine the modeling performance. The overall evaluation of datapoints not only deciphers the distribution in various aspects (e.g., composition, task, and domain) but also helps cherry-pick the most beneficial subsets for higher performance with less training cost. Through this survey, we demonstrate that: 1) existing resourceful data assessment methods can be categorized into three main perspectives: quality, diversity, and importance (see Fig. 1). 2) a systematic view of selection methods can be unified even they more or less exhibit coupling with the assessment techniques (see Fig. 2). It is noted that quality, diversity, and importance might be used interchangeably without strict discrimination in previous studies. But here we provide a rationalized organization taxonomy for structured elaboration. Despite the goal of being comprehensive, the present survey only provides details of certain typical, representative methods to avoid being tediously long. We hope the in-depth explanations and discussions on the selected methods provide insights into developing robust data assessment and selection pipelines for future studies.

## 1.1 Related Surveys

[Liu et al. (2024d)] studied the mainstream datasets for building LLMs, including the pre-training corpora, instruction tuning datasets, preference datasets, evaluation benchmarks, and traditional NLP datasets. Their work focuses on the descriptions of dataset statistics (e.g., categorization, sources, and domains) without providing guidelines on utilization. In contrast, we emphasize the selection of instruction-tuning data for the improved downstream performance. [Albalak et al. (2024)] presented a systematic overview of constructing the data pipeline for language models. Any selection method, either via distribution matching or diversification, can be composed of: 1) utility function; 2) selection mechanism. During different stages of the pipeline, the selection method should be adjusted according to different selection objectives (e.g., language filtering, data quality control , domain knowledge division , deduplication, toxic and explicit content removal, and data mixing). Their work pays extra attention to the processing of the pre-training corpora while neglecting the fine-grained analysis of existing selection methods specifically designed for instruction tuning. In this case, our survey serves as an indispensable extension on the selection of instruction datasets. [Wang et al. (2024a)] focused on the data preparation for instruction tuning. Existing methods on building instruction tuning datasets include: 1) reformulating the discriminative NLP datasets into generative ones; 2) self-instruct with seed prompts; 3) prompt mapping and evol-instruct Popular methods on dataset selection can be simply classified as: 1) system of indicators; 2) trainable LLMs; 3) powerful LLMs; and 4) small models.

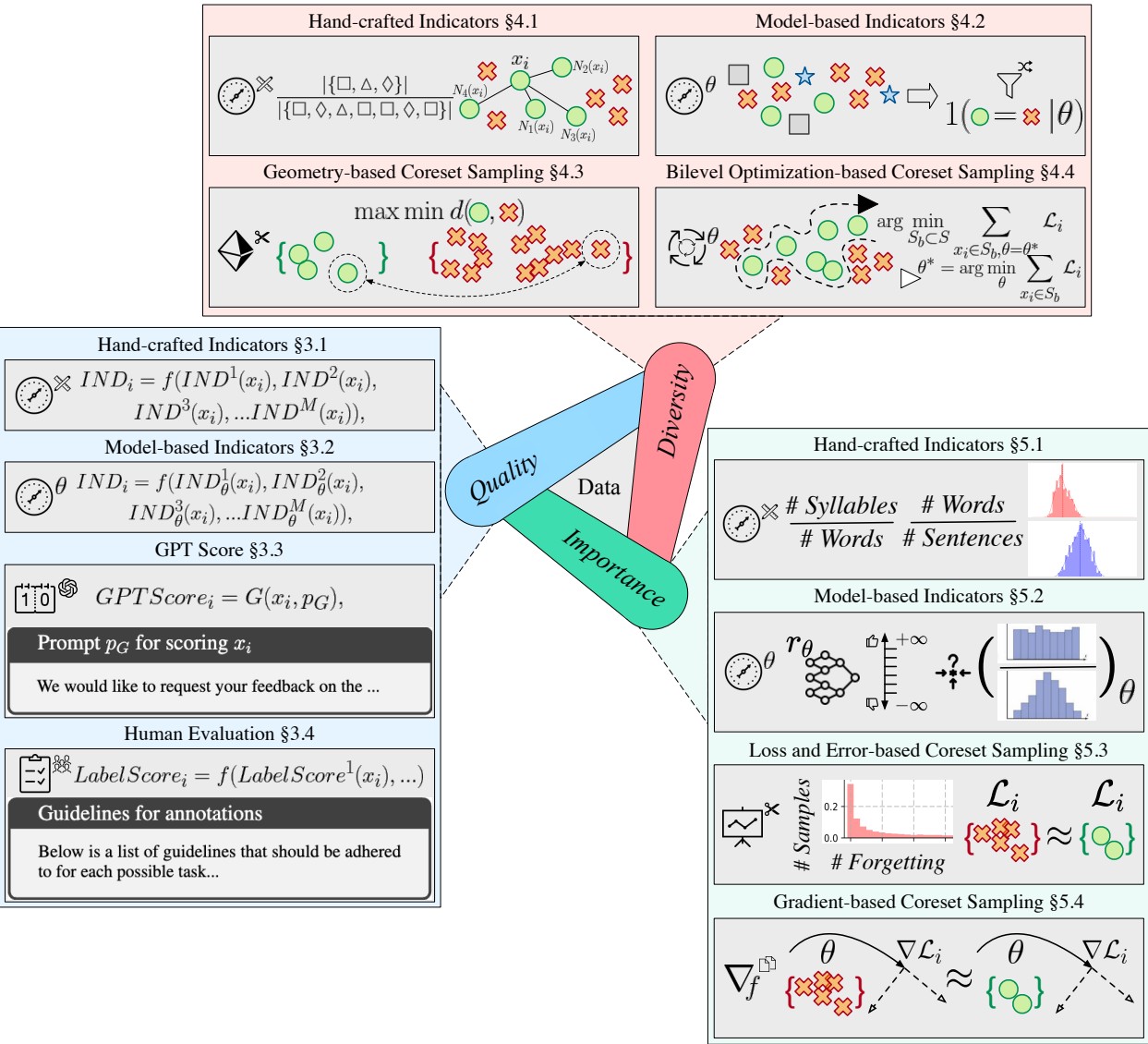

Figure 1: Categorization of data assessment and selection methods for effective instruction tuning of LLMs.

Comparatively, our survey stems from the characteristics of data themselves, namely quality, diversity, and importance, for categorization of selection methods. In each category, we further provide subdivided groups by the selection philosophy, which deepens the understanding of selection for practical take-home messages.

Existing surveys on general data selection also shed light on the principles of developing selection methods. [Guo et al. (2022)] started from the general coreset selection methods in the field of deep learning and categorize all selection manners into: 1) geometry-based methods (e.g., herding, k-center greedy); 2) uncertainty-based methods (e.g., least confidence/entropy/margin); 3) error/loss-based methods (forgetting; GraND/EL2N; importance resampling); 4) decision boundary-based (adversarial deepfool; contrastive active learning); 5) gradient matching-based (gradient approximation towards full set); 6) bi-level optimization-based (inner loop of model optimization and outer loop of datapoint selection); 7) sub-modularity-based (e.g., graph cut; facility location); 8) proxy-based (preference of a small model on data selection). [Zhou et al. (2024b)] investigated the potential metrics and aspects for data quality measurement. They provide a list of available tools for data evaluation. Apart from data assessment and selection methods that are specifically designed for NLP or

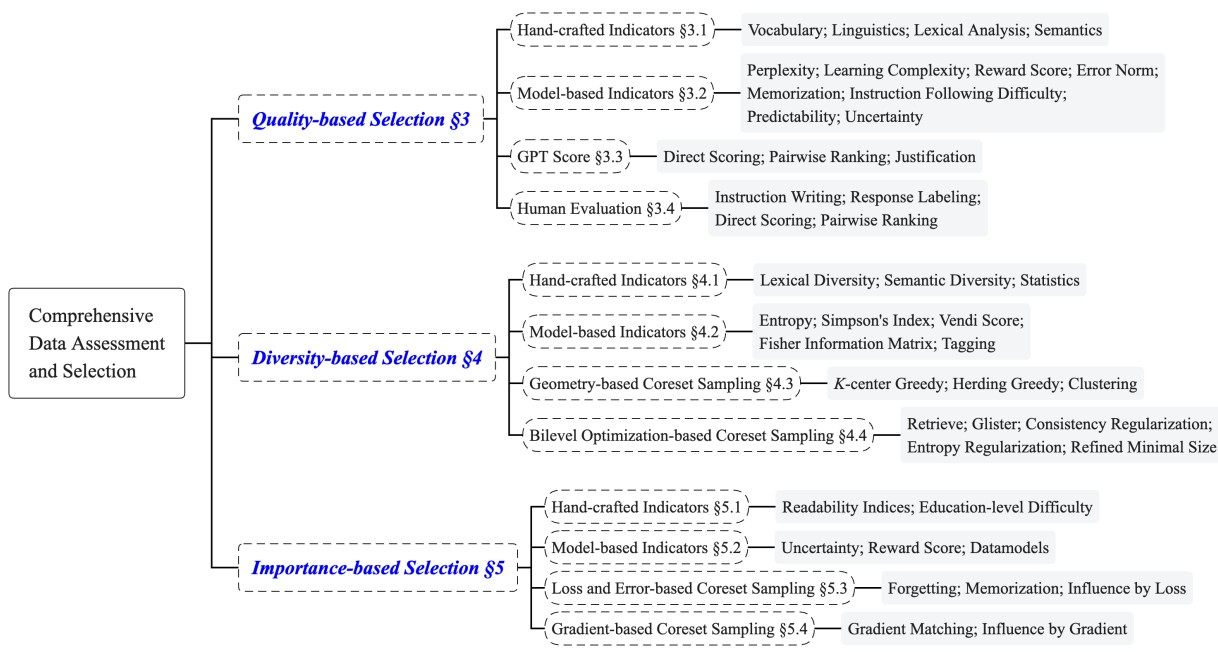

Figure 2: A high-level overview of comprehensive data assessment and selection. The analysis aspects that apply to either individuals or the overall dataset can be categorized into three groups marked in blue italic.

LLM applications [Moore & Lewis (2010); Chen et al. (2024a); Dodge et al. (2020); Kandpal et al. (2022); Li et al. (2022); Feng et al. (2021); Lee et al. (2021); Malhotra & Bakal (2015); Liu et al. (2024e)], there exist many survey studies that tackle general quality measurement in machine learning [Gupta et al. (2021); Zha et al. (2023); Ehrlinger & Wöß (2022); Mohammed et al. (2024); Li et al. (2024c); Lu et al. (2023b); Dix et al. (2023); Priestley et al. (2023); Byabazaire et al. (2020); Roh et al. (2019); Sidi et al. (2012); Batini et al. (2009)] for constructing safe, unbiased, and accurate datasets. In this paper, we mainly focus on reviewing data selection methods for utilizing instruction tuning data in the context of large language models.

## 1.2 Survey Scope

Although "data evaluation" has been so frequently mentioned that it appears as a cliché problem in developing machine learning algorithms, the optimal solution to establishing an overall data assessment and selection pipeline still remains an open question. Especially under the context of instruction tuning of LLMs, existing studies proposed various measurements and strategies to select the "high-quality" instructions. However, very few studies noticed that there exists no unified dimensions or aspects in measuring data "quality" where previous works tend to put emphasis on the domain-specific and task-dependent characteristics. In addition, the inherent, systematic coupling between data assessment and subset selection is not well demonstrated.

Under such circumstance, the present study strives to provide a comprehensive review on evaluating and decomposing massive instruction tuning datasets. We categorize the main aspects of data assessment in terms of **quality**, **diversity**, and **importance**. To reduce ambiguity, their definitions are first provided below.

Quality refers to the intrinsic value of the data. High-quality data typically satisfy two conditions: 1) The instructions are clear, accurate, and explicit in explaining the task at hand and the expected behavior of LLMs. 2) The responses are correct, coherent, and pertinent to the instructions. All the requirements and constraints specified in instructions should be met. Accordingly, most existing methods not only underline the clarity and fluency of instructions but also filter out mismatched, incorrect, and harmful responses via various judging and measuring techniques.

Diversity refers to the variety and richness of the dataset. During training, models that are exposed to datapoints under a wide range of scenarios enjoy a high level of generalization to unseen tasks. High-diversity data imply that each constituting datapoint is unique. The diverse dataset should cover different knowledge domains and task definitions with varying input contexts and output constraints. In light of this statement, existing methods that prioritize diversity all emphasize the removal of homogeneous, near-deduplicated datapoints in terms of lexical and semantic representations.

Importance refers to the impact of specific data points on the LLM's performance. It implies the necessity of adding one datapoint into the selected subset for instruction tuning. The importance of a datapoint is up to factors such as the difficulty of following the instruction, the capability of the LLM under investigation, and the scale of the dataset to be selected. Under such circumstance, various methods of importance-based selection have been developed by measuring the difficulty of each datapoint, matching the gradients or performance of models that are trained on the selected and the full set, and investigating the dynamics of models in learning rich and scarce datasets.

In each aspect, we provide a detailed survey on both traditional (e.g., hand-crafted indicators) and machine learning (e.g., model-based indicators) methods. Besides, the coreset sampling methods that bridge evaluation and selection are introduced separately in diversity and importance oriented subset construction. In consideration of the properties of instruction tuning, we focus on the text modality and start from classical text analysis metrics. Metrics that are either specific to instruction tuning or compatible with pre-training and preference alignment are included since they all share general rules in data assessment.

The survey is organized as follows. First, we present the preliminaries for assessment and selection of instruction tuning datasets (Sec. §2). Next, we present the surveying methods of data assessment and selection methods in terms of quality (Sec. §3), diversity (Sec. §4), and importance (Sec. §5). Then, discussions on the existing methods are provided in (Sec. §6), followed by the promising directions for future research (Sec. §7). The final conclusion is given in (Sec. §8).

### 1.3 What's Beyond the Survey Scope

The instruction tuning methods for de-bias and fairness of LLMs are not covered in the present study. We acknowledge that the bias and fairness aspects of instruction data are valued in developing responsible LLMs [Gallegos et al. (2024); Chu et al. (2024); Li et al. (2023d)]. However, they are beyond the scope of the present study for the following reasons:

- Most **data selection** methods on instruction tuning do not even notice the bias or fairness of data. They are in lack of explicitly designing steps to reduce negative impacts of biased data.

- Existing fairness tuning methods follow their own definitions of quality and diversity in terms of gender, race, religion, profession, age, and political ideology. It is difficult to bring in all their corresponding evaluation techniques under our selection taxonomy.

- The bias and fairness not only intersect with diversity but also with quality control, which is not an inclusive concept. For example, to control the harmful contents like hatred and racism responses, reward models or GPT scoring from quality-based selection can be used because the publicly released reward models and LLMs are already aligned with human preference.

- The evaluation of bias and fairness is not covered in existing studies on data selection, which makes it difficult to validate the effectiveness of selection techniques in improving the fairness of LLMs.

However, we emphasize that bias and fairness should be specifically handled because instruction-tuned LLMs tend to exhibit more bias than the pre-trained LLMs [Itzhak et al. (2024)]. Therefore, we believe the selection for a less-biased dataset would be a promising future direction in developing comprehensive selection methods.

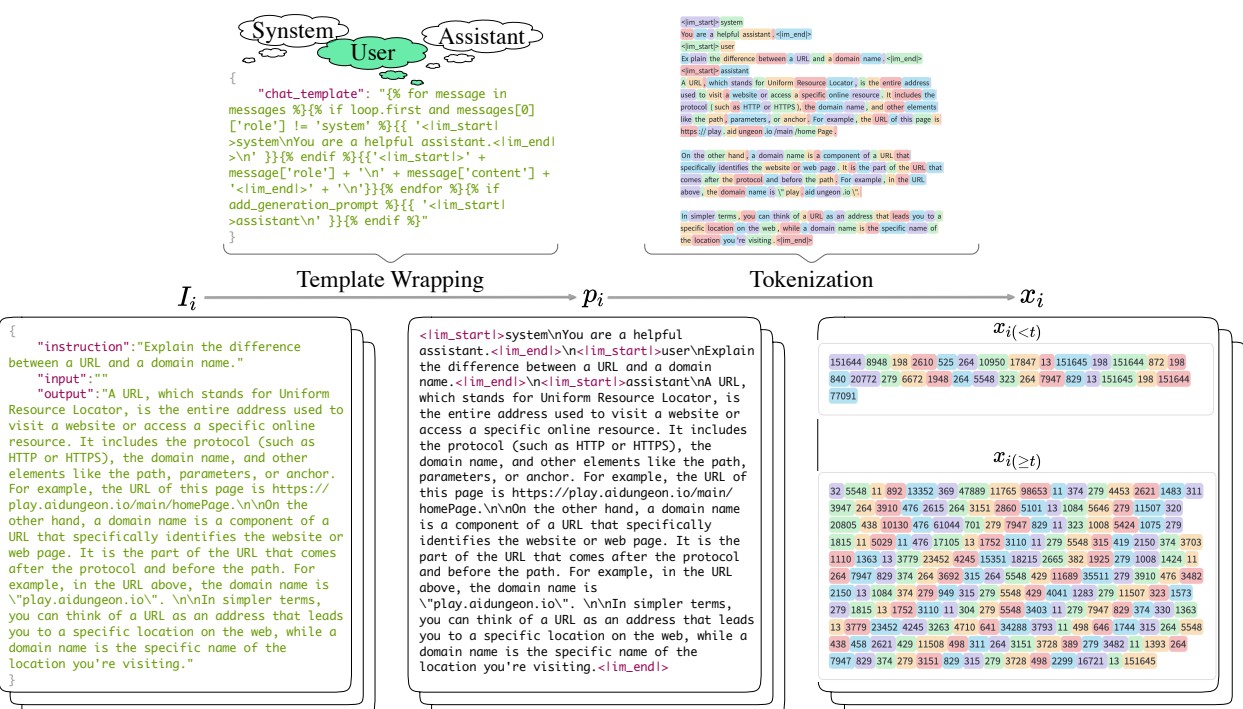

Figure 3: The pre-processing of an instruction dataset includes: 1) template wrapping, and 2) tokenization. In the first step, we wrap the raw texts $I_i$ with a pre-defined chat template into the textual prompts $p_i$. In the second step, we perform tokenization on $p_i$ with the LLM-associated tokenizer for the datapoint $x_i$. Given the index $t$ for indicating where the loss mask of language modeling starts taking effect, we split $x_i$ into $x_{i(<t)}$ and $x_{i(\geq t)}$, respectively denoting the instruction part (input) and the response part (output).

## 2 Preliminaries

In this section, we briefly introduce the instruction tuning of LLMs and the problem statement for dataset assessment and selection.

**Instruction Dataset Preparation** In instruction tuning, each text sample $I_i$ is usually composed of three parts: 1) instruction (either with or without system prompt), 2) input (can be empty), and 3) response.

For an off-the-shelf pre-trained LLM parameterized as $\theta$, a pre-determined instruction template is used to wrap the $I_i$ into a textual prompt $p_i$ with special tokens like "<|im_start|>" and "<|im_end|>" for separation of roles (e.g., system, user, assistant, function, and observation) and their contents. It is noted that the prompt template for organizing a text sample $I_i$ differs across model families. Especially, the special tokens are unique to the tokenizers and play an important role in differentiating multi-turn conversations and stopping model generations. Then, a LLM-associated tokenizer performs tokenization on the textual prompt $p_i$ for a sequence of token-id integers: $x_i = [x_{i(1)}, x_{i(2)}, ..., x_{i(n)}]$, where $x_{i(j)}$ denotes the $j$-th token of $x_i$ and $n$ is the total number of tokens. Out of simplicity, the token sequence $x_i$ can be simply split into two parts by the index $t$: 1) the instruction part ($x_{i(<t)}$) that is fed into the LLM without being involved in loss computation, and 2) the ground-truth response part ($x_{i(\geq t)}$) that is expected as the LLM output via minimizing its language modeling loss. Without losing generality, we use the terminology of "datapoints" throughout the paper to refer to the tokenized samples $x_i$.

The process of processing an instruction sample $x_i$ is illustrated in Fig. 3. We choose one example from the Alpaca dataset [Taori et al. (2023)] and perform Qwen tokenization [Yang et al. (2024)] for demonstration.

**Instruction Supervision**   Given the tokenized instruction tuning dataset $S = \{x_i\}_{i=1}^{N}$, the supervised tuning is performed via cross-entropy loss:

$$\mathcal{L} = \sum_{x_i \in S} \mathcal{L}_i,$$

$$\mathcal{L}_i = -\sum_{j=t}^{|x_i|} \log P(x_{i(j)}|x_{i(<j)}; \theta). \tag{1}$$

For each $x_i$, given all previous tokens $x_{i(<j)}$ , the model iteratively predicts the next token $x_{i(j)}$ at the $j$-th index. Note that the previous tokens here include both the instruction context part $x_{i(<t)}$ and the response completions up to the current token $x_{i(\geq t, <j)}$ .

**Data Assessment and Selection**   We aim at finding the most informative subset $S_b \subset S$ from the entire set $S$ under the given budget $|S_b| \leq b$. Mathematically, the selection of $S_b$ requires: 1) a quantitative evaluation function $q(\cdot)$ that assesses each datapoint $x_i$, and 2) an elaborated sampling mechanism $\pi$ that determines the rules of selection:

$$S_b = \pi(S, b, q). \tag{2}$$

With respect to the detailed implementation of $\pi$, either an iterative, greedy algorithm or a batch-wise heuristic rule can be adopted for compatibility with $q(\cdot)$. For example, the greedy sampling is represented as:

$$S_b = \pi_{\text{greedy}}(S, b, q) = \arg \max_{S' \subseteq S, |S'| \leq b} \sum_{x_i \in S'} q(x_i). \tag{3}$$

The greedy algorithm $\pi_{\text{greedy}}$ consistently selects datapoint $x_i$ with the highest score $q(x_i)$ until the budget target $b$ is met. The heuristic sampling algorithm, on the other hand, chooses each datapoint $x_i$ according to a pre-defined rule. For example, the probability derived by the score $q(x_i)$ can be employed as reference:

$$S_b = \pi_{\text{prob}}(S, b, q) = \text{Sample}(S, b, p),$$

$$p(x_i) = \frac{q(x_i)}{\sum_{x_j \in S} q(x_j)}, \tag{4}$$

where $\text{Sample}(S, b, p)$ performs sampling $b$ times upon the normalized probability distribution over $q(\cdot)$. More advanced sampling techniques $\pi$ can be developed in accordance with the domains and tasks at hand.

Such data assessment and selection paradigm is expected to bring about the following benefits:   1) the reduction of noise by ignoring those mislabeled, mismatched instruction-response pairs, 2) the re-balance of data distributions by down-sampling those easy, common, and similar examples while up-sampling hard, rare, and unique  ones, and 3) the expedition of training in return for efficient optimization  of LLMs.

## 3   Quality-based Selection

In this section, we present methods on quality assessment and selection. Without lose of generality, we present the unified formulation of quality measurement. The evaluation function $q(x_i)$ in quality-based methods can be decomposed into two fine-grained parts: 1) instruction quality $q_I(x_i)$ and 2) response quality $q_R(x_i)$.

$$q(x_i) = f_q(q_I(x_{i<t}), q_R(x_{i \geq t})), \tag{5}$$

where $f_q$ is an aggregation function that combines the instruction and response quality scores either explicitly or implicitly. Specifically, the instruction quality $q_I$ can be further broken down into: 1) clarity $q_I^C$ that measures the ease of understanding the task, 2) accuracy $q_I^A$ that measures how well the instruction aligns with the expected task, and 3) explicitness $q_I^E$ that measures how explicitly the instruction defines the output constraints (e.g., formats and styles). Consequently, we have $q_I(x_{i<t}) = g_I(q_I^C(x_{i<t}), q_I^A(x_{i<t}), q_I^E(x_{i<t}))$ with the aggregation function $g_I$. Similarly for the measurement of response, its quality $q_R$ can be assessed via: 1) correctness $q_R^C$ that measures whether the response correctly answers the instruction, 2) coherence $q_R^H$ that

measures the logical consistency of the response, and 3) pertinence $q_R^P$ that measures the relevance of the response to the instruction. The final response can be judged as $q_R(x_{i \geq t}) = g_R(q_R^C(x_{i<t}), q_R^H(x_{i<t}), q_R^P(x_{i<t}))$ with an aggregation function $g_R$. It is noted that all the mentioned quality measurement components above are only demonstrative and are not enforced explicitly in the development of existing quality-based methods. Therefore, certain fine-grained aspects might be integrated into one formulation without extra considerations.

### 3.1 Hand-crafted Indicators

**Overview**  Traditional methods develop hand-crafted indicators to evaluate the data quality in terms of linguistic analysis such as vocabulary, syntax, and inter-sample semantic similarity. Each indicator is manually, empirically designed with prior knowledge on the language, domain, and task of the corpus under investigation. The calculation of each indicator is explicitly defined and does not require training and inference of proxy models or language models. Although the indicators are hand-crafted, deep learning models such as sentence encoders might be leveraged to extract embedding representations for each datapoint $x_i$. An indicator $IND_i$ can be typically defined as:

$$
\begin{aligned}
IND_i = f(IND^1(x_i), IND^2(x_i), \\
IND^3(x_i), ...IND^M(x_i)),
\end{aligned}
\tag{6}
$$

where $M$ denotes the total number of indicators and $f$ is the aggregation function which depends on both the instruction task and dataset. One can simply use linear combinations with pre-defined or dynamically adjusted weights. However, meticulous tuning might be needed for the ultimate $f$. Given the indicators $IND_i$ for each $x_i$, two intuitive selection methods can be adopted: 1) to filter out datapoints whose indicator scores are below a threshold; 2) to keep only the samples whose indicator scores rank within a certain range of percentiles. Mathematically, these two selection mechanisms can be respectively represented as:

$$
S_\pi = \{x_i | \tau_{\min} < f(x_i) < \tau_{\max}, 1 \leq i \leq N\},
\tag{7}
$$

$$
S_\pi = \{x_i | P_{\min} \leq \hat{F}_f(f(x_i)) \leq P_{\max}, 1 \leq i \leq N\},
\tag{8}
$$

where $\tau_{\min}$ and $\tau_{\max}$ respectively denote the left and right threshold boundaries. The estimated $\hat{F}_f$ is the empirical cumulative distribution function of all indicators $f$. $P_{\min}$ and $P_{\max}$ respectively refer to the minimum and maximum percentile for enclosing the selection range. In practice, both the threshold and percentiles are hyper-parameters that require task-specific fine-tuning.

**Technical Details**  [Mishra et al. (2020a)] and [Mishra et al. (2020b)] introduce a data quality metric, namely the DQI, to quantify the differences between successive benchmarks by giving high scores to generalizable samples and low scores to biased samples. Such a metric implies whether a well-trained model truly learns the underlying task rather than overfitting the spurious bias of specific benchmarks. Specifically, DQI has seven components including vocabulary, inter-sample N-gram frequency and relation, inter-sample semantic textual similarities (STS), intra-sample word similarity, intra-sample STS, N-Gram frequency per label, and inter-split STS. Based on the DQI, [Mishra & Sachdeva (2020)] propose to prune existing huge NLP datasets and demonstrates that the model trained on only 2% of the SNLI dataset achieves near-equal performance with that on the entire set. It first performs AFLite [Le Bras et al. (2020)], which is detailed in [Sakaguchi et al. (2021)], to keep samples with predictability scores over a threshold and then delete bottom $k$ samples with the lowest DQI scores. [Dang & Verma (2024)] further split DQI components into linguistic indicators and semantic indicators, and validate their respective roles in detecting outliers, noises, and duplications. Apart from data selection for training LLMs, quality indicators can also be employed to identify the most discriminative samples in the evaluation set to expedite evaluation of LLMs. [Saranathan et al.] investigate key indicators such as spelling errors [Yannakoudakis & Fawthrop (1983)], average word length, excessive word repetition, and the compound probability distribution. These indicators stem from the traditional studies on text readability (i.e., readability formulas and sophisticated features) [Klare et al. (1963; 1984); Dubay (2004); Kintsch & Vipond (2014); Kemper (1983)]. Recent studies on readability leverage NLP systems to extract more advanced and informative features for readability measures [Si & Callan (2001); Collins-Thompson & Callan (2005); Schwarm & Ostendorf (2005); Feng et al. (2010)]. [François (2010; 2011); François & Fairon

(2012)] systematically analyze the lexical features, syntactic features, semantic features, and language-specific features with up to 46 indicators. [François & Miltsakaki (2012)] validate these manually-designed (classical) and NLP-enabled (non-classical) readability formulas, implying that high-quality texts can be pinpointed by such carefully designed metrics. [Felice & Specia (2012)] find that the hand-crafted linguistic features should be combined with other shallow features for better quality estimation.

**Remark**   The hand-crafted indicators often stem from studies on linguistic analysis and readability measurement. Although these indicators help filter out instruction samples that are unreadable, nonsensical, and incoherent, they cannot detect mismatched instruction-response pairs and therefore fail to guarantee the instruction-following capability of LLMs trained on highly-scored datasets.

### 3.2   Model-based Indicators

**Overview**   The model-based indicators, on the other hand, leverage trainable models to predict the indicators for each datapoint. The trainable models used for data quality measurement can either share the same or similar architecture with the language model under development, or possess completely different implementations. Accordingly, these indicators can be simply defined as:

$$IND_i = f(IND^1_{\theta_1}(x_i), IND^2_{\theta_2}(x_i),$$
$$IND^3_{\theta_3}(x_i), ...IND^M_{\theta_M}(x_i)), \tag{9}$$

where the learnable parameters $\theta_1$, $\theta_2$, ..., $\theta_M$ highlight the difference between model-based and hand-crafted indicators. Based on such  indicators, similar selection mechanisms (Eqs. 7 and 8) can be adopted.

**Technical Details**   One of the most intuitive model-based indicators is perplexity [Shannon (2001); Jelinek et al. (1977); Jelinek (1980)]. It is frequently mentioned as the evaluation metric for pre-trained language models [Penedo et al. (2023); Radford et al. (2018; 2019); Brown et al. (2020); Achiam et al. (2023)] but can also be employed as a data quality indicator. [Ankner et al. (2024)] propose to use a small GPT-style reference model such as MPT 125M [Team (2023)] to prune datasets via perplexity-based sampling for training a 3B model. For any datapoint $x_i$, the perplexity is defined as the exponential of negative likelihood with base of 2:

$$NLL_i = \frac{1}{|x_i|} \sum_{j=1}^{|x_i|} -\log P(x_{i(j)}|x_{i(<j)}; \theta) \tag{10}$$
$$PPLX_i = 2^{NLL_i}$$

Based on the perplexity inferred from a small model, samples at the high and medium percentiles are chosen by Eq. 8 for downstream fine-tuning.  [Deng et al. (2021)] develop a unified evaluator framework to score the generated outputs for natural language generation tasks. A RoBERTa-based [Liu et al. (2019)] discriminator learns to score responses in terms of consistency, relevance, preservation, engagingness, and groundedness. One could simply adopt such a discriminator for evaluation of instruction-response pairs. [Zhong et al. (2022)] further propose a multi-dimensional scoring evaluator. For each evaluation dimension, the original instruction-response pairs are converted into positive samples in the form of boolean question-answer problems. The negative samples are respectively constructed via a rule-based transformation. The evaluator itself is implemented as a T5 model [Raffel et al. (2020)] and trained on these positive and negative samples for scoring in the range from 0 to 1.  [Jiang et al. (2024c)] prune the UltraChat [Ding et al. (2023)] dataset by scoring each datapoint by the learning complexity of a pre-trained Qwen-1.8B model [Bai et al. (2023)]. Specifically, the learning complexity is calculated as the averaged prediction confidence of different subnets:

$$\tilde{S}(x_i) = \frac{1}{I} \sum_{j=1}^{I} PPLX_{i;\Theta_j}, \tag{11}$$

where $I$ is the number of subnets. Each subnet $\Theta_j$ is obtained by adjusting the dropout rate from 10% to 90% on the original $\Theta$ of any pre-trained language model. Such a dropout technique imitates the learning

progress in a training-free manner. Datapoints with small $\tilde{S}(x_i)$ are considered as easy ones. They are often learned at an earlier training stage with less model capacity compared with hard ones. In a data-poor regime, easy datapoints are more informative and should be kept first. On the contrary, in a data-rich regime, hard datapoints should be treasured in that more samples can be preserved for fine-tuning. Both [Bukharin & Zhao (2023)] and [Du et al. (2023)] employ reward models to assess the quality of each instruction pairs. They respectively utilize the raft model [Dong et al. (2023)] and the deberta-v3-large-v2 [1] for reward scoring:

$$R_i = r_\theta(x_{i(<t)}, x_{i(\geq t)}), \tag{12}$$

where $r_\theta$ denotes the reward model. $t$ is the index where $x_{i(<t)}$ and $x_{i(\geq t)}$ respectively denote the instruction $Q$ and response $A$. [Marion et al. (2023)] investigate three classic metrics in clean set selection [Guo et al. (2022); Song et al. (2022); Natarajan et al. (2013); Qin et al. (2024)]: perplexity (Eq. 10), error $l_2$-Norm (EL2N) [Paul et al. (2021)], and memorization ranking [Biderman et al. (2024)]. Specifically, EL2N is defined as:

$$EL2N_i = \frac{1}{|x_i|} \sum_{j=1}^{|x_i|} \|P(x_{i(<j)}; \theta) - \mathbf{y}_{i(j)}\|_2, \tag{13}$$

where $\mathbf{y}_{i(j)}$ denotes the one-hot ground-truth vector as the target of the probability vector $P(x_{i(<j)}; \theta)$. Specifically, both $P(x_{i(<j)}; \theta) \in \mathbb{R}^{N_{\text{vocab}}}$ and $\mathbf{y}_{i(j)} \in \mathbb{R}^{N_{\text{vocab}}}$ are of the same dimension, where $N_{\text{vocab}}$ denotes the vocabulary size associated with the tokenizer. For the vector of $\mathbf{y}_{i(j)}$, all elements are zero except that the element indexed at $x_{i(j)}$ is one. The memorization ranking is represented as:

$$MEM_i = \frac{1}{N_{\text{win}}} \sum_{j=1}^{N_{\text{win}}} \mathbb{1}(\hat{x}_{i(M_{\text{offset}}+j)} = x_{i(M_{\text{offset}}+j)}), \tag{14}$$

where $N_{\text{win}}$ denotes the length of a consecutive sequence and $M_{\text{offset}}$ is an offset of the starting index. The $\hat{x}_{i(M_{\text{offset}}+j)}$ refers to the generated token given input $x_{i(<M_{\text{offset}}+j)}$, and $x_{i(M_{\text{offset}}+j)}$ is its ground-truth. [Cao et al. (2023)] combine both hand-crafted indicators (e.g., input length, output length, MTLD [McCarthy & Jarvis (2010)], and kNN-i [Dong et al. (2011)]) and model-based indicators (e.g., reward score, perplexity, and Uni-Eval metrics [Zhong et al. (2022)]) for fitting the loss of a LLM on the evaluation set. The linear regression model is optimized via least squares method [Bjork (1988)] and the optimal selection of instruction data is achieved via BlendSearch [Wang et al. (2021a;b)] for minimizing the estimated evaluation loss. [Li et al. (2023a)] propose one of the most pioneering works that leverages the target language model itself to perform self-guided data selection. The language model is first "warmed-up" with very few samples randomly chosen from the pool to learn from brief experience. Then, such an experienced model evaluates each instruction-response pair via the instruction-following difficulty (IFD) score. The IFD score measures how much guidance or assistance the instruction provides to the generation of ground-truth response, by comparing the loss of causal language modeling on the response with and without instruction:

$$IFD_i = \frac{NLL_i^{A|Q}}{NLL_i^A},$$

$$NLL_i^{A|Q} = \frac{1}{|x_{i(\geq t)}|} \sum_{j=t}^{|x_i|} -\log P(x_{i(j)}|x_{i(<j)}; \theta), \tag{15}$$

$$NLL_i^A = \frac{1}{|x_{i(\geq t)}|} \sum_{j=t}^{|x_i|} -\log P(x_{i(j)}|x_{i(t\leq, <j)}; \theta),$$

where the index $t$ splits apart the instruction $Q$ and the response $A$. Samples whose IFD scores over $\tau_{\max} = 1$ are invalid datapoints with misaligned, mismatched instruction-response pairs. The empirical setting of $\tau_{\min}$ affects the trade-off between quality and diversity of the selected datapoints. [Zhao & Fang (2024)] comprehensively employ hand-crafted indicators for low-level quality filtering, and uses perplexity and IFD score for high-level filtering. A voting mechanism is additionally introduced with IFD scores from one

---

[1]https://huggingface.co/OpenAssistant/reward-model-deberta-v3-large-v2

pre-trained base model and one fine-tuned experience model. [Li et al. (2024b)] corroborate that both the perplexity and IFD scores inferred from a rather small GPT2-125M [Radford et al. (2019)] are indicative in selecting high-quality datapoints for training LLaMA2-7B and LLaMA2-13B [Touvron et al. (2023b)], which greatly improves selection efficiency.

Another popular model-based quality filtering method is AF-Lite [Le Bras et al. (2020)], which has been applied and validated in recent NLP studies [Mishra & Sachdeva (2020); Sakaguchi et al. (2021)]. It randomly partition all available datapoints into the training set and the validation set. Then, a model (e.g., linear classifier or language model) is trained on the training set and inferred on the validation set. Such process iterates $m$ times for calculation of the predictability score, which is defined as the ratio of the number of correctly predicted response over the number of total predictions:

$$PRED_i = \frac{|\{\hat{x}_i \in E_i, \ s.t. \ \hat{x}_i = x_i\}|}{E_i},$$
$$E_i = \{\hat{x}_i^{\theta_1}, \hat{x}_i^{\theta_1}, ..., \hat{x}_i^{\theta_j}, ..., \hat{x}_i^{\theta_m}\}, \tag{16}$$

where $\hat{x}_i^{\theta_j}$ denotes the generated response from the model parameterized as $\theta_j$. It is noted that $x_i$ is not involved for optimizing $\theta_j$, and therefore a higher $PRED_i$ suggests better quality.

[Bhatt et al. (2024)] present uncertainty-based indicators such as mean entropy [Settles (2011); Kremer et al. (2014)], least confidence [Settles (1995; 2011)], mean margin [Tong & Koller (2001); Balcan et al. (2006); Settles (2011)], and min margin [Nguyen et al. (2022)]. Mathematically, such uncertainty indicators are defined as:

$$U_i^{\text{entropy}} = \frac{1}{|x_i|} \sum_{j=1}^{|x_i|} P(x_{i(j)}|x_{i(<j)};\theta) \cdot$$
$$\log P(x_{i(j)}|x_{i(<j)};\theta). \tag{17}$$

$$U_i^{\text{confidence}} = -\prod_{j=1}^{|x_i|} P(x_{i(j)}|x_{i(<j)};\theta). \tag{18}$$

$$U_i^{\text{margin}} = -\frac{1}{|x_i|} \sum_{j=1}^{|x_i|} (\beta_1(P(x_{i(<j)};\theta)) -$$
$$\beta_2(P(x_{i(<j)};\theta))), \tag{19}$$

$$U_i^{\text{min-margin}} = -\min_{j \in \{1,2,...,|x_i|\}} (\beta_1(P(x_{i(<j)};\theta)) -$$
$$\beta_2(P(x_{i(<j)};\theta))), \tag{20}$$

where $\beta_1$ and $\beta_2$ denote the largest and second largest elements of the probability $P(x_{i(<j)};\theta) \in \mathbb{R}^{N_{\text{vocab}}}$ for the newly generated $j$-th token. However, [Wu et al. (2023)] find that such uncertainty-based data sampling methods perform worse than random sampling on Databricks-Dolly [Conover et al. (2023)], SelfInstruct-Davinci [Taori et al. (2023)], and SelfInstruct-GPT4 [Peng et al. (2023)].

**Remark** Hybrid techniques that simultaneously combine perplexity, uncertainty, reward scores, and other training-aware metrics are promising in selecting unbiased high quality samples. In consideration of the training and inference cost, it is feasible to employ small proxy models as alternatives for computing model-based indicators.

### 3.3 GPT Score

**Overview** The invoking of OpenAI APIs [Tingiris & Kinsella (2021); Lappalainen & Narayanan (2023); Sun et al. (2023); Kublik & Saboo (2023)] for ChatGPT services (e.g., GPT3.5, GPT4) allows automatic scoring of instruction tuning datasets. Recent studies on bringing LLMs as judges [Zheng et al. (2024); Wang et al. (2023a); Zhu et al. (2023); Huang et al. (2024); Zeng et al. (2023); Chan et al. (2023); Pang et al.

(2024)] reveal that powerful language models like ChatGPT highly align with human preference on judging the quality of instructions and responses. Given a well-designed prompt with clear definition on grading criteria, ChatGPT produces justified quality scorings with explanations for the raw textual instruction data $I_i$:

$$GPTScore_i = G(I_i, p_G), \tag{21}$$

where $p_G$ denotes the prompt template that defines the task and grading scheme with format constraints on outputs (see Fig. 4). The $G(\cdot, \cdot)$ represents the process of quality scoring and response parsing, which produces the GPT score $GPTScore_i$. Samples with high $GPTScore_i$ can be selected using Eqs. 7 and 8.

---

**Prompt $p_G$ for scoring $I_i$ with instruction (input) and response in the \<dimension\>**

We would like to request your feedback on the performance of AI assistant in response to the instruction and the given input displayed following.

Instruction: \<instruction\>
Input: \<input\>
Response: \<response\>

Please rate according to the \<dimension\> of the response to the instruction and the input. Each assistant receives a score on a scale of 0 to 5, where a higher score indicates higher level of the \<dimension\>. Please first output a single line containing the value indicating the scores. In the subsequent line, please provide a comprehensive explanation of your evaluation, avoiding any potential bias.

---

Figure 4: The prompt $p_G$ for scoring the raw text $I_i$ with ChatGPT.

**Technical Details** [Chen et al. (2023b)] propose a surprisingly easy-yet-effective method that directly uses GPT3.5 to score datapoints in terms of helpfulness and accuracy. Both instructions and responses are scored on a scale from 0 to 5 and experimental results show that general instruction datasets, except coding-related samples, can be distilled into smaller subsets for better downstream performance. [Bukharin & Zhao (2023)] follow [Chen et al. (2023b)] for filtering Alpaca [Taori et al. (2023)]. [Chen & Mueller (2024)] employ the BSDetector [Chen & Mueller (2023)] to estimate the confidence of GPT3.5/GPT4 on the give instruction-response pair. It takes both the self-consistency and direct scoring into consideration. Only highly confident samples are kept for fine-tuning domain-specific LLMs and those less confident ones are corrected automatically by these LLMs. [Xu et al. (2023b)] directly evaluate instruction datasets in terms of accuracy, explanation, clarity, and difficulty for weighted scorings from GPT4. Then, both hand-crafted indicators (i.e., lengthwise semantic evaluation) and GPT4 scorings are employed for final ranking. [Liu et al. (2023b)] argue that the direct scoring of GPT4 on one single instruction sample is not well-calibrated and instead gives relative ranking of multiple instruction variants at once. The complexity of instructions [Xu et al. (2023a)] and the quality of instruction-response pairs are sequentially obtained from GPT3.5. [Zhang et al. (2024c)] use GPT scorings to judge: 1) whether the given text contains mathematical contents; 2) and if yes, whether these maths contents are of high quality for education purpose. Such scores are proved more effective than traditional "mathematical" classifiers [Paster et al. (2023)]. [Lu et al. (2023a)] propose to use ChatGPT for annotating open-ended, fine-grained intention tags on open datasets. Then, the quality of the tag dataset is evaluated by humans and GPT4 in terms of tagging precision and consistency. Instead of fully relying on the GPT4, [Li et al. (2023c)] exploit the model under investigation itself (e.g., LLaMA 65B) to iteratively derive quality scores on each augmented example on a 5-point scale. Then a curated clean set is chosen via the Eq. 7. QuRator [Wettig et al. (2024)] manually define quality criterion such as writing style, facts and trivia, educational value, and required expertise. Then, quality comparison is conducted on two instruction-response samples via GPT3.5 scoring. Such pairwise scorings are used to fine-tune a sheared-LLaMA 1.3B model [Xia et al. (2023)] in a manner similar to DPO [Ouyang et al. (2022); Rafailov et al. (2024)]. It is noted that the pairwise scoring [Ouyang et al. (2022); Dubois et al. (2024); Zeng et al. (2023); Liu et al. (2023b)] has been found more reliable, consistent, and unbiased than the individual scoring [Gunasekar et al. (2023); Chen et al. (2023b)] during GPT-based quality analysis.

**Remark** Closed-source LLMs such as ChatGPT enjoy a high level of alignment with human preference and therefore can be utilized for quality scoring. It would be cost-efficient to collect few (e.g., <100K) GPT-scored samples first and then fine-tune an open-source LLM for quality measurement on massive corpus.

## 3.4 Human Evaluation

**Overview** Evaluation with human-in-the-loop is indispensable in constructing preference alignment datasets [Wang et al. (2023b); Ouyang et al. (2022)] for helpfulness, honesty, and harmlessness. Specifically, human annotators deliver gradings following specific criteria (see Fig. 5) in multiple dimensions:

$$LabelScore_i = f(LabelScore^1(x_i), LabelScore^2(x_i), ..., LabelScore^M(x_i)), \tag{22}$$

where $LabelScore^m(x_i)$ can be both bool or integer (e.g., range from 0 to 5) for the $m$-th fine-grained aspect. The aggregation function $f$ is commonly chosen as summation or averaging.

---

**Guidelines (excerpts) for human annotations**

\# Guidelines
Below is a list of guidelines that should be adhered to for each possible task available when building the dataset. To see some examples of how the guidelines can be applied, visit the examples document.

\#\# 1. General rules
- Always make sure to read and understand the guidelines to each task before fulfilling it. - Try to follow the guidelines as closely as possible. - If you are unsure whether a message violates a guidelines, contact us at our Discord.
- Use the thumbs-up/thumbs-down system to further mark messages that are of high or low quality.

\#\# 2. Providing an assistant reply #assistant-reply
\#\#\# Do:
- Remain polite and treat the user with respect, even when not given the same courtesy.
...

---

Figure 5: The guidelines (thumbnails) for human experts to create and annotate instruction datasets.

**Technical Details** The OpenAssistant [Köpf et al. (2024)] dataset is featured by its high-quality human-generated, human-annotated multi-lingual conversations for both instruction tuning and reinforcement learning from human feedback. For each instruction-response pair along the conversation tree, the human annotators are asked to categorize them according to three dimensions: spam detection, guideline adherence, and quality. The quality score is rated on a five-point *Likert* scale across aspects including quality, creativity, humorousness, politeness, and harmlessness. These scores are used to sort instructions for analysis and preference optimization of LLMs. [Lu et al. (2023a)] enroll human annotators to provide judgements on the tagging of each instruction. To verify the quality scores provided by humans, counterfactual cases are prepared respectively for precision and consistency tasks. Results show that human annotators have low false positive rates at tagging precision, but lack proof of confidence on their original quality judgements. [Zhou et al. (2024a)] propose to use human annotators for creation of small-yet-effective instruction datasets. To collect questions and answers from various sources, simple hand-crafted indicators such as text length are used to filter low-quality datapoints. Then, high quality instruction-response pairs are manually selected (750) and written (250) via subjective quality control. The databricks-dolly dataset [Conover et al. (2023)] contains 15K human-generated instruction-response pairs. Although quality is emphasized during large-scale annotation, imperfect samples still exist. For example, low-quality and inaccurate responses, incomplete and vague instructions, problematic texts with toxic language and grammar errors are found [He et al. (2024)].

**Remark** Human evaluation plays an irreplaceable role in quality control of preference alignment. To reduce the inter-annotator inconsistency, detailed guidelines should be prepared for quality measurement. In addition,

supplementary quality measures such as GPT scores can be provided for reference during evaluation and selection of high-quality datapoints.

# 4 Diversity-based Selection

In this section, we introduce methods that emphasize the diversity of instruction datasets. When it comes to diversity, existing researches either measure the individual diversity of each sample (e.g., lexical and semantic richness) or the overall diversity of the entire dataset (e.g., the volume of the enclosed embedding space). Datapoints whose tasks and domains are of minority classes in a long-tailed distribution are preferred during subset selection. Such sampling philosophy strikes to maintain or approximate the spread of the original embedding clusters but with much less sparsity. In the case of diversity, the evaluation function $q(x_i)$ can be represented in the unified formulation as:

$$q(x_i) = f_d(q_L(x_i), q_S(x_i)), \tag{23}$$

where $q_L$ measures the lexical diversity of $x_i$ and $q_S$ assesses the semantic diversity. The $f_d$ denotes the aggregation function in diversity measurement. Typically, $q_L$ often investigates the diversity of n-grams, tokens, words, and sequences. Complementarily, $q_S$ emphasizes semantic diversity that the variety of representations of the selected datapoints should be maximized in the embedding space. Both the two aspects of diversity can be sequentially or jointly considered to remove any duplicates in the instruction datasets.

## 4.1 Hand-crafted Indicators

**Overview**   The diversity of datasets is the key to develop less biased, more generalizable machine learning models. However, recent studies [Zhao et al. (2024c;b)] show that existing datasets do not share a unified and concrete definition of diversity in terms of dataset composition, source, domain, subject, annotator, and promote (fairness). With respect to the diversity measures specific in instruction tuning datasets, hand-crafted indicators, similar to Eq. 6 in traditional NLP studies, can be used as a good starting point.

**Technical Details**   One of the most popular diversity measure is lexical diversity, which refers to the range of different words occurring in one text. The greater range implies greater diversity and quality. Type-token ratio (TTR) [Templin (1957); Richards (1987)] is originally proposed as:

$$TTR_i = \frac{|Unique(x_i)|}{|x_i|}, \tag{24}$$

where $Unique(x_i)$ denotes the set of unique tokens present in $x_i$. To reduce the sensitivity of TTR to the variation of text length, several studies [Covington & McFall (2008; 2010); Kettunen (2014); Matlach et al. (2021)] standardized the length by introducing logarithms or n-grams into the formula.

Later, computational approaches to measure lexical diversity have been developed such as vocabulary diversity (vocd-D) [Malvern & Richards (1997); Malvern et al. (2004); Silverman & Ratner (2002); deBoer (2014)], the measure of textual lexical diversity (MTLD) [McCarthy & Jarvis (2010); Jarvis & Daller (2013)], and hypergeometric distribution diversity (HD-D) [Jarvis (2013); McCarthy (2005)]. All these metrics require multi-step computation for approximation. Specifically for vocd-D, random sampling is first performed on $x_i$ for a series of sub-sequences with varying lengths $k$ (e.g., 10, 20, 30 tokens). Then, $TTR^k$ is:

$$TTR_i^k = \frac{|Unique(x_{i(j \leq, <j+k))}|}{|x_{i(j \leq, <j+k)}|}, \ 1 \leq j \leq |x_i| - k, \tag{25}$$

where $x_{i(j \leq, <j+k))}$ denotes the sub-sequence of $x_i$ starting from the randomly chosen index $j$ and ending at the index $j + k$. Then, the curve of $TTR_i^k$ versus the lengths $k$ is plotted and a mathematical model is built for fitting the curve:

$$\hat{TTR}_i^k = \frac{\mathcal{D}}{k}[(1 + 2\frac{k}{\mathcal{D}})^{\frac{1}{2}} - 1], \tag{26}$$

where $\mathcal{D}$ is the only parameter required to be estimated. By approximating $T\hat{T}R_i^k$ towards $TTR_i^k$ with the least squares, we have $\mathcal{D}_{\text{best fit}} = D$:

$$vocd\text{-}D_i = D. \tag{27}$$

A larger $D$ reflects the higher diversity of $x_i$. The computation of MTLD, on the other hand, first determines the $TTR_i$ as a pre-defined threshold, and then partitions $x_i$ into $M$ different contiguous subsequences $\{x_i^1, x_i^2, ..., x_i^m, ..., x_i^M\}$. Each subsequence $x_i^m = x_{i(j \leq, < j+k)}, \forall k > 0, \forall 1 \leq j \leq |x_i| - k$ maintains a $TTR_i^k$ above the threshold $TTR_i$. The MTLD is defined as:

$$MTLD_i = \frac{1}{M} \sum_{i=1}^{M} |x_i^m|. \tag{28}$$

The HD-D shares the same idea behind vocd-D but stems from the hypergeometric distribution [McCarthy & Jarvis (2010)]. With $M$-times sampling, the HD-D represents the probability of drawing a certain number of tokens of the given type from the subsequence of $x_i$ with a particular size $k$:

$$HD\text{-}D_i = \sum_{t=1}^{|Unique(x_i)|} \frac{1}{M} \sum_{m=1}^{M} \mathbb{1}(x_{i(n)}^m = u_t), 1 \leq n \leq |x_i^m|, \tag{29}$$
$$u_t \in Unique(x_i), x_i^m = x_{i(j \leq, < j+k)}, \forall k > 0, \forall 1 \leq j \leq |x_i| - k.$$

Variants of TTR indicators such as MTTRSS [Malvern et al. (2004)], MSTTR [Malvern et al. (2004)], MATTR [Covington & McFall (2010)], and MTLD-W [Vidal & Jarvis (2020); Kyle et al. (2021)] all target at the solutions to two fundamental problems [Bestgen (2023)]: 1) the sensitivity of indicators to text length, and 2) the impact of the indicator parameters. [Li et al. (2015)] propose two rather simplified TTR scores as *distinct*-1 and *distinct*-2, where the number of distinct unigrams and bigrams of $x_i$ are respectively divided by the total number of tokens. Other studies [Cao & Clark (2017); Zhu et al. (2018); Shu et al. (2019); Tevet & Berant (2020)] extend the application of n-gram-based diversity for model-generated responses.

Apart from lexical diversity, there exists many efficient diversity indicators that are built upon the semantics of each example. [Dong et al. (2011)] propose to approximate k-nearest neighbor (k-NN) graph [Peterson (2009)] with arbitrary similarity measures on semantic embeddings of large-scale datasets. Such efficient construction of a k-NN graph allows the distance of $x_i$ to its $j$-th nearest neighbors to be a feasible diversity measure:

$$kNN_i^j = d(g(x_i), g(N_j(x_i))), \tag{30}$$

where $N_j(x_i)$ denotes the $j$-th closest neighbor of $x_i$ in the embedding space projected by $g(\cdot)$. The common choices of the distance function $d(\cdot, \cdot)$ include the Euclidean distance, cosine distance, and Jaccard coefficient distance [Huang et al. (2008)]. The projection from text (e.g., instruction-response pairs) into the embedding space can be achieved with pre-trained sentence BERT [Reimers & Gurevych (2019); Feng et al. (2020)], where an additional pooling operation is performed on the final output of BERT [Devlin et al. (2018)] for sentence embeddings. Note that a higher $kNN_i$ implies that the sample $x_i$ is more unique and should be kept in subset selection for higher diversity. Due to the fine-grained representation capability of BERT, existing hand-crafted indicators often rely on BERT embeddings for similarity or diversity measurement [Tevet & Berant (2020); Zhang et al. (2019); Larson et al. (2019); Yauney et al. (2023)].

To improve the generalization of diversity measure, [Xu et al. (2023b)] argue that the statistics of feature embedding of each sample itself should be considered. It does not require additional prior knowledge on the structure of embeddings. Given all datapoints $x_i \in S$, their semantic embeddings from any sentence encoder can be represented as $X = [g(x_1), g(x_2), ..., g(x_N)] \in \mathcal{R}^{|S| \times H}$. The row variance $Var_i$ of each embedding $g(x_i)$ in the reduced dimensional space $\mathcal{R}^{|S| \times k}$ by principal components analysis (PCA) [Wold et al. (1987)] is used as the diversity indicator:

$$Var_i = \frac{1}{k-1} \sum (j=1)^k (Y_{ij} - \mu_i)^2,$$
$$\mu_i = \frac{1}{k} \sum_{j=1}^{k} Y_{ij} \tag{31}$$

where the PCA chooses the top-k eigenvectors ($V = [v_1, v_2, ..., v_k]$ with $\lambda_1 \geq \lambda_2 \geq ... \geq \lambda_k$) of the covariance matrix $Cov = Q\Lambda Q^T = \frac{1}{|S|-1}(X - \mu_X)^T(X - \mu_X)$, $\mu_X = \frac{1}{|S|}\sum_{i=1}^{|S|} X_i$ to project the original embeddings into more compact and reduced ones via $Y = (X - \mu_X)V$. Samples with the highest 20% $Var_i$ (via Eq. 8) are selected as the variety-curated dataset.

When it comes to the overall diversity of a dataset $S$, the average distance of any datapoint $x_i$ to its closest neighbor in the dataset, namely $kNN_i$, can be leveraged intuitively:

$$D^{kNN}(S) = \frac{1}{|S|}\sum_{i=1}^{|S|} kNN_i^1, \ x_i \in S. \tag{32}$$

Such a diversity measure has been widely used in dataset construction and retrieval [Stasaski et al. (2020); Stasaski & Hearst (2022); Mithun et al. (2019); Spyromitros-Xioufis et al. (2015); Sun et al. (2024a); Ionescu et al. (2018)]. [Du & Black (2019)] simply perform clustering on all samples with k-means [Ikotun et al. (2023)] into $K$ clusters ($C_1, C_2, ..., C_K$) in the embedding space, and then uses the cluster inertia as diversity indicators:

$$D^{inertia}(S) = \sum_{j=1}^{K} \sum_{x_i \in C_j} \|g(x_i) - \mu_j\|^2,$$
$$\mu_j = \frac{1}{|C_j|} \sum_{x_i \in C_j} g(x_i). \tag{33}$$

[Lai et al. (2020)] develop a diversity metric on the dispersion of a cluster induced by embeddings of all samples, where the cluster is approximated by a multi-variate Gaussian distribution:

$$D^{radius}(S) = \sqrt[H]{\prod_{j=1}^{H} \sigma_j}, \tag{34}$$

where $H$ is the dimension of the projected embeddings $g(x_i) \in \mathcal{R}^H$ and $\sigma_j$ denotes the radius of the ellipsoid along the $j$-th axis of the dataset $S$. The inter-cluster (class) distance can also be used for diversity measure [Dang & Verma (2024)]:

$$D^{ICD}(S) = \frac{1}{K}\sum_{j=1}^{K} \text{div}_{JS}(P_j || P_{\neq j}), \tag{35}$$

where $P_j$ denotes the inverse-document frequency (IDF) distribution [Sparck Jones (1972)] of the cluster $C_j$ and $\text{div}_{JS}$ is the Jensen-Shannon divergence.

**Remark** Both lexical and semantic diversity should be considered with hand-crafted indicators. The optimization of individual diversity would contribute to the overall diversity of the entire dataset.

### 4.2 Model-based Indicators

**Overview** Similar to Eq. 9, model-based indicators on diversity also rely on the target or proxy language model for computing the indices.

**Technical Details** The diversity of a dataset $S$ can be intuitively defined as the sum of rarity measures of each constituting element $x_i$. Accordingly, entropy-related methods are proposed to estimate such rarity. The more uncommon, various samples exist, the higher diversity the dataset becomes. Mathematically, the vanilla entropy [Shannon (1948)] is proposed for diversity measures:

$$D^{entropy}(S) = -\sum_{x_i \in S} P(x_i|\theta) \cdot \log_2(P(x_i|\theta)), \tag{36}$$

where $P(x_i)$ denotes the probability of $x_i$ occurring in the dataset. Later, Rényi entropy [Rényi (1961)] introduce an additional parameter $\alpha > 0, \alpha \neq 1$ for a generalized entropy definition:

$$D_\alpha^{RE}(S) = \frac{1}{1-\alpha} \log_2(\sum_{x_i \in S} P(x_i|\theta)^\alpha). \tag{37}$$

The parameter $\alpha$ adjusts the element-wise emphasis on rare or frequent events.

Studies on biology and ecology [Mouillot & Lepretre (1999); Peet (1974); He & Hu (2005); Gregorius & Gillet (2008)] investigate Simpson's Index (SI) [Simpson (1949); Wu et al. (2024; 2022)] for measuring the biodiversity of species and genetics. [Zhou et al. (2020)] propose a variant of the original SI with a more flexible statistic metric:

$$D^{SI}(S) = 2 \frac{\sum_{x_i,x_j \in S, i \leq j} \mathbb{1}(x_i = x_j|\theta)}{|S|(|S|+1)}, \tag{38}$$

where the equivalence of $x_i$ and $x_j$ is judged by an indicator function parameterized as $\theta$.

Vendi Score (VS) [Dan Friedman & Dieng (2023); Pasarkar & Dieng (2023); Nguyen & Dieng (2024)] is rencently proposed for diversity measurement in machine learning researches. Inspired by the Rényi entropy, a generalized VS metric [Pasarkar & Dieng (2023)] is defined as below:

$$D_\alpha^{VS}(S) = \exp(\frac{1}{1-\alpha} \log_2(\sum_{i=1, i \in supp(\bar\lambda)}^{|S|} \bar\lambda_{i|\theta}^\alpha)), \tag{39}$$

where $\bar\lambda_{i|\theta}$ denotes the normalized eigenvalues of the similarity kernel matrix $K_{S|\theta}$, and $supp(\bar\lambda)$ is the set of indices of all non-zero eigenvalues. The smaller $\alpha < 1$ makes the scoring more sensitive to rare classes and therefore allows accurate diversity measurement even under severe class imbalance. One simple implementation of the similarity kernel $K_{S|\theta}$ is to use the Gaussian Radial Basis function $k$ with feature embeddings as $k(g(x_i|\theta), g(x_j|\theta)) = \exp(-\frac{1}{2}\|g(x_i|\theta) - g(x_j|\theta)\|^2)$. [Nguyen & Dieng (2024)] further introduce quality scoring into Eq. 39 where for each subset $S_b \subset S$, its average quality score $Q(S_b) = \frac{1}{|S_b|} \sum_{x_i \in S_b} IND_i$ is multiplied with $D_\alpha^{VS}(S_b)$ for comprehensive evaluation in terms of quality and diversity.

[Miranda et al. (2022)] propose an intrinsic diversity coefficient to measure the diversity of a dataset with Task2Vec embeddings [Achille et al. (2019); Nguyen et al. (2019)] for distance computation between different tasks. The Task2Vec encodes data from different tasks by the diagonal entries of the Fisher Information Matrix (FIM). The FIM results from fine-tuning only the final (e.g., token classification) layer of a pre-trained model, namely a probe model (e.g., GPT2 [Radford et al. (2019)]), to solve the task. Given a batch of samples $B$, the mathematical representation of FIM is defined as:

$$\hat{F}_B = \mathbb{E}_{x_i,j,\hat{x}_{i(j)}} \nabla_\theta \log P(\hat{x}_{i(j)}|x_{i(<j)};\theta) \cdot$$
$$\nabla_\theta \log P(\hat{x}_{i(j)}|x_{i(<j)};\theta)^T, \tag{40}$$

where $\hat{x}_{i(j)}$ denotes the $j$-th token predicted from the model parameterized as $\theta$ given the real sequence input $x_{i(<j)}$. The expectation $\mathbb{E}_{x_i,j,\hat{x}_{i(j)}}$ takes an average over the sequence length $|x_i|$ for each $x_i$ sampled randomly from the batch $x_i \in B$. The Task2Vec embedding $\vec{f}_B = diag(F_B)$, where $diag(\cdot)$ denotes the diagonal entries of $F_B$. Based on the Task2Vec embeddings, [Lee et al. (2023)] propose to compute the diversity coefficients $\hat{div}$ specifically for NLP datasets:

$$D^{\hat{div}}(S) = \mathbb{E}_{B_1,B_2 \sim S} d(\vec{f}_{B1}, \vec{f}_{B2}),$$
$$D^{\hat{div}}(S_1, S_2) = \mathbb{E}_{B_1 \sim S_1, B_2 \sim S_2} d(\vec{f}_{B1}, \vec{f}_{B2}), \tag{41}$$

where $d$ denotes distance measurement (e.g., cosine distance). Both $B_1$ and $B_2$ are two batches sampled respectively from the same or different datasets for diversity measures within or across datasets. Experiments confirm that hand-crafted indicators such as the number of latent concepts [Xie et al. (2021)] and the richness of vocabulary are positively associated with the proposed $\hat{div}$ coefficients.

[Lu et al. (2023a)] develop a diversity measure by open-ended tagging. Specifically, a tagging model parameterized by $\theta$ is trained with GPT4-labeled tagging pairs. To label such tags of each datapoint, the GPT4 first performs open-set fine-grained tagging and then all the collected tags are normalized to filter out low-frequency ones and aggregate near-duplicate ones. The specifically trained tagging model describes each instruction tuning datapoint $x_i$ by its fine-grained, atomic intentions and semantics (e.g., tasks and domains). Correspondingly, the number of tags can be viewed as a diversity indicator for sampling a instruction subset $S_b$ from the whole set $S$ (see Alg. 1).

---

**Algorithm 1** TagLM-based Diverse Sampling [Lu et al. (2023a)]

---

**Require:** data $x_i \in S$, a tagging LLM $T_\theta$, a visited tag set $D_b^B$, and a budget $b$
 1: Initialize $S_b = \emptyset$
 2: **for** each $x_i \in S$ **do**
 3:     Obtain tags $D_{x_i} = T_\theta(x_i)$
 4: **end for**
 5: **repeat**
 6:     Initialize $D_b^B = \emptyset$
 7:     **for** each $x_i = \arg\max_{x_i \in S} D_{x_i}$ **do**
 8:         **if** $|D_b^B \cup D_{x_i}| > |D_b^B|$ **then**
 9:             $S_b = S_b \cup \{x_i\}$
10:             $D_b^B = D_b^B \cup D_{x_i}$
11:             $S = S \backslash \{x_i\}$
12:         **end if**
13:     **end for**
14: **until** $|S_b| = b$
15: **return** $S_b$

---

**Remark** The model-based indicators are highlighted by their flexibility in handling various aspects of diversity either implicitly or explicitly.

## 4.3 Geometry-based Coreset Sampling

**Overview** Instead of explicitly calculating the diversity-aware indicators, recent studies on selecting instruction datasets tend to introduce coreset sampling methods for a systematic consideration [Guo et al. (2022)]. Specifically, coreset sampling aims to find the most informative-and-diverse subset that represents the entire dataset the most, so that close or even surpassing performance can be achieved on the language model trained on the subset with respect to that on the entire set.

**Technical Details** Among different categories of coreset sampling methods, geometry-based methods are the most intuitive and widely-used ones [Chen et al. (2012); Agarwal et al. (2020); Sener & Savarese (2017); Sinha et al. (2020); Kamalov (2020); Rezazadegan Tavakoli et al. (2011); Kirchenbauer et al. (2024); Zhou et al. (2023)]. The intuition behind is that close samples in the embedding space often share similar properties with low diversity. Therefore, redundant information can be effectively suppressed by controlling the minimum distance between any two samples for subset selection. Specifically, k-center greedy is a typical diversity-oriented sampling method for massive pretraining and instruction-tuning corpus [Chen et al. (2023a); Bhatt et al. (2024); Wu et al. (2023); Zhao & Fang (2024); Du et al. (2023)]. It solves the minimax facility location (FL) problem [Cornuéjols et al. (1983); Farahani & Hekmatfar (2009)], i.e., selecting the subset $S_b$ under the given size budget $b$ from the full set $S$ so that the largest distance between an example in $S \backslash S_b$ and its closest example in $S_b$ is minimized:

$$\min_{S_b \subset S, \ |S_b| = b} \max_{x_i \in S \backslash S_b} \min_{x_j \in S_b} d(g(x_i), g(x_j)). \tag{42}$$

The direct solution to Eq. 42 is NP-hard [Cook et al. (1994)] and a greedy approximation is proposed [Sener & Savarese (2017)] (see Alg. 2). Instead of simultaneously retrieving all the datapoints that can maximize

the diversity of the selected subset, k-center greedy iteratively finds the most heterogeneous datapoint until the budget $b$ runs out. For initialization of $S_b^0$, one can either choose randomly sampled datapoints from $S$, or use the cluster center points from $K$ clusters $(C_1, C_2, ..., C_K)$ of $S$ via k-means clustering. Similarly, the farthest point sampling method [Eldar et al. (1997)] shares the same principle that each iteration time only the farthest datapoint relative to the already selected coreset is chosen from the candidates.

---

**Algorithm 2** K-Center Greedy [Sener & Savarese (2017)]

---

**Require:** data $x_i \in S$, existing pool $S_b^0$ and a budget $b$
 1: Initialize $S_b = S_b^0$
 2: **repeat**
 3:     $u = \arg\max_{x_i \in S \setminus S_b}$
                 $\min_{x_j \in S_b} d(g(x_i), g(x_j))$
 4:     $S_b = S_b \cup \{u\}$
 5: **until** $|S_b| = b + |S_b^0|$
 6: **return** $S_b \setminus S_b^0$

---

In addition to the k-center greedy, the herding methods [Chen et al. (2012); Welling (2009); Huszár & Duvenaud (2012); Adhikary & Boots (2022)] select datapoints $x_i$ so that the distance between the coreset center and the full set center is minimized in the embedding space. For efficiency, it is also approximated via greedy implementation [Chen et al. (2016); Harvey & Samadi (2014)] by adding one sample each time into the $S_b$ to minimize the distance between two centers (see Alg. 3). Similar to k-center greedy, the herding greedy also performs iterative sample selection. However, it differs in that in each step, the distance between the centers of the selected set and the full set is minimized.

---

**Algorithm 3** Herding Greedy [Harvey & Samadi (2014)]

---

**Require:** data $x_i \in S$, a budget $b$
 1: Initialize $\mu = \frac{1}{n} \sum_{i=1}^n g(x_i)$
 2: Initialize $S_b = \emptyset$
 3: **for** $t = 1$ to $b$ **do**
 4:     $u = \arg\min_{x_i \in S \setminus S_b} \|\mu -$
         $\frac{1}{|S_b|+1} \sum_{x_j \in S_b \cup \{x_i\}} g(x_j)\|^2$
 5:     $S_b = S_b \cup \{u\}$
 6: **end for**
 7: **return** $S_b$

---

Furthermore, recent studies tend to develop complex heuristic sampling methods that takes geometry-based diversity into consideration [Jiang et al. (2023c); Chan et al. (2021); Xia et al. (2022)]. Specifically, the inter-sample similarity of the selected coreset is minimized in return for an overall high diversity. [Jiang et al. (2024c)] propose to preserve informative subset with the learning complexity (see Eq. 11) and implicitly puts constraints on its diversity via sampling on the k-means clusters:

$$D^{dist}(S) = \frac{1}{|S|} \sum_{x_i \in S} \min_{j \neq i} d(x_i, x_j) \geq C, \tag{43}$$

where $C$ denotes the constant that controls the degree of diversity. A larger $C$ represents the larger diversity of the dataset. The detailed procedure can be found in Alg. 4. It can be seen that datapoints are first sampled cluster-by-cluster to ensure a more balanced data distribution. Then, a percentile-based selection is performed on each cluster to select datapoints with easy learning complexity.

[Bukharin & Zhao (2023)] propose the quality-diversity instruction tuning (QDIT). It also uses FL functions for diversity measure of the subset $S_b$:

$$D^{FL}(S_b) = \sum_{x_j \in S} \max_{x_i \in S_b} sim(g(x_i), g(x_j)), \tag{44}$$

---

**Algorithm 4** Easy and Diverse First Sampling [Jiang et al. (2024c)]

---

**Require:** data $x_i \in S$, existing pool $S_b^0$, a budget $b$, and the number of clusters $K$
1: Initialize $S_b = S_b^0$
2: $\arg\min_C \sum_{j=1}^{K} \sum_{x_i \in C_j \subset S} \|\frac{g(x_i)}{\|g(x_i)\|} - \mu_j\|^2$,
    $\mu_j = \frac{1}{|C_j|} \sum_{x_i \in C_j} \frac{g(x_i)}{\|g(x_i)\|}$.
3: **for** $j = 1$ to $K$ **do**
4:     $S_b^j = \{x_i | \hat{F}_{\tilde{S}}(\tilde{S}(x_i)) \leq \frac{b}{K}, x_i \in C_j\}$
5:     $S_b = S_b \cup S_b^j$
6: **end for**
7: **return** $S_b$

---

where $sim(\cdot, \cdot)$ denotes the similarity function (e.g., cosine similarity). If the selected $S_b$ can be well-representative of the entire set $S$, then $S_b$ is assumed of high diversity. Given quality scores defined by Eq. 21, the detailed mechanism of QDIT is described in Alg. 5 with greedy approximation. Different from the previous sequential setup that respectively prioritizes quality and diversity in two successive steps, QDIT adopts a dynamic weighting over the GPT scoring-based quality and the FL-based diversity for selection.

---

**Algorithm 5** QDIT sampling [Bukharin & Zhao (2023)]

---

**Require:** data $x_i \in S$, a budget $b$, and the trade-off hyper-parameter $\alpha$
1: Initialize $S_b = \emptyset$
2: **for** $t = 1$ to $b$ **do**
3:     $u = \arg\max_{x_i \in S \setminus S_b} (1 - \alpha) \cdot D^{FL}(S_b \cup \{x_i\}) + \alpha \cdot GPTScore_i$
4:     $S_b = S_b \cup \{u\}$
5: **end for**
6: **return** $S_b$

---

[Liu et al. (2023b)] adopt the quality score-first and diversity-aware data selection method (DEITA), where all datapoints are first scored and sorted by quality measurement, and then selected by a geometry-based heuristic criterion (i.e., Repr Filter). Specifically, it considers that for each chosen datapoint in $S_b$, its $kNN_i^1$ (Eq. 30) should be above a certain threshold $\tau$ so that the overall diversity $D^{kNN}(S_b)$ (Eq. 32) can be improved. As shown in Alg. 6, the quality and complexity of each sample $x_i$ are respectively measured by the trained complexity scoring model $\theta_C$ and the quality scoring model $\theta_Q$ with prompts $p_C$ and $p_Q$. Then, samples with high $G_{CQ}$ are prioritized but only those dissimilar ones can be kept for the diversity of $S_b$.

---

**Algorithm 6** DEITA Sampling [Liu et al. (2023b)]

---

**Require:** data $x_i \in S$ and a budget $b$
1: Compute the combined complexity and quality score $G_{CQ}(x_i) = G(x_i, p_C | \theta_C) \cdot G(x_i, p_Q | \theta_Q)$
2: $u = \arg\max_{x_i \in S} G_{CQ}(x_i | \theta)$
3: Initialize $S_b = \{u\}$
4: $S = S \setminus \{u\}$
5: **while** $|S_b| < b$ **do**
6:     $u = \arg\max_{x_i \in S} G_{CQ}(x_i | \theta)$
7:     **if** $d(g(u), g(N_0(u))) > \tau, N_0(u) \in S_b$ **then**
8:         $S_b = S_b \cup \{u\}$
9:     **end if**
10:     $S = S \setminus \{u\}$
11: **end while**
12: **return** $S_b$

---

Another series of geometry-based methods focus on the organization of data structures via developing clustering-based sampling techniques [Citovsky et al. (2021); Tirumala et al. (2024); Axiotis et al. (2024);

Shao et al. (2024); Alcoforado et al. (2024); Saranathan et al.]. With respect to the clustering criterion, traditional methods employ topic modeling with LDA [Blei et al. (2003); Raghuveer et al. (2012); Bui et al. (2017)], NMF [Lee & Seung (2000); Wang & Zhang (2012); Shen & Si (2010); Lazar & Doncescu (2009)], TF-IDF [Sparck Jones (1972); Bafna et al. (2016); Patil & Atique (2013); Roul et al. (2014)], and latent concepts [Xie et al. (2021)] to assign text corpus into thematic clusters. Most recent studies exploit sentence encoding methods [Reimers & Gurevych (2019); Feng et al. (2020)] to perform clustering in the embedding space, where the vanilla k-means clustering and its variants [Sinaga & Yang (2020); Kanungo et al. (2000); Bandyapadhyay & Varadarajan (2015)], DBSCAN [Deng (2020); Khan et al. (2014); Crețulescu et al. (2019)], and spectral clustering [Bach & Jordan (2003); Von Luxburg (2007); Jia et al. (2014)] are widely used. Specifically, [Tirumala et al. (2024)] propose to use SemDeDup [Abbas et al. (2023)] to remove semantically similar examples for deduplication, which provides a basis of diversity sampling. Then, k-means clustering is performed in the embedding space. In each cluster, the prototype-based sampling technique [Sorscher et al. (2022)] is used. The "prototypical" samples, whose distance to their assigned cluster centers are small, should be discarded first to allow more "outliers" to be kept in $S_b$ during iterative sampling (see Alg. 7). It is noted that two times of clustering operations are conducted, where the semantic deduplication and prototype-based sampling are respectively performed on $K_1$ and $K_2$ clusters.

---

**Algorithm 7** D4 Sampling [Liu et al. (2023b)]

---

**Require:** data $x_i \in S$, a budget $b$, the number of clusters for SemDeDup $K_1$ and the number of clusters for prototypicality $K_2$

1: Initialize $S_d = \emptyset, S_b = \emptyset$

2: $\arg\min_C \sum_{j=1}^{K_1} \sum_{x_i \in C_j \subset S} \| \frac{g(x_i)}{\|g(x_i)\|} - \mu_j \|^2$,
   $\mu_j = \frac{1}{|C_j|} \sum_{x_i \in C_j} \frac{g(x_i)}{\|g(x_i)\|}$.

3: **for** $j = 1$ to $K_1$ **do**

4:  $C_j^v = \emptyset$

5:  **while** $|C_j^v| < |C_j|$ **do**

6:    $u = \arg\min_{x_i \in C_j \setminus C_j^v} sim(g(x_i), \mu_j)$

7:    **if** $\max_{x_i \in C_j} sim(g(u), g(x_i)) < \tau$ **then**

8:      $S_d = S_d \cup \{u\}$

9:    **end if**

10:   $C_j^v = C_j^v \cup \{u\}$

11:  **end while**

12: **end for**

13: $\arg\min_C \sum_{j=1}^{K_2} \sum_{x_i \in C_j \subset S_d} \| \frac{g(x_i)}{\|g(x_i)\|} - \mu_j \|^2$,
   $\mu_j = \frac{1}{|C_j|} \sum_{x_i \in C_j} \frac{g(x_i)}{\|g(x_i)\|}$.

14: **for** $j = 1$ to $K_2$ **do**

15:  $S_b^j = \{x_i | \hat{F}_d(d(x_i, \mu_j) > \frac{b}{K_2}, x_i \in C_j\}$

16:  $S_b = S_b \cup S_b^j$

17: **end for**

18: **return** $S_b$

---

[Axiotis et al. (2024)] propose a k-means cluster-based sensitivity sampling technique. For each datapoint in a cluster, both its distance to the cluster center and a proxy evaluation loss [Feldman & Langberg (2011)] measured on that cluster center contribute proportional to the probability of being chosen. [Shao et al. (2024)] propose the balanced ClusterClip sampling. It first performs k-means clustering and then sample datapoints uniformly from each cluster. Different from the uniform sampling, the proposed ClusterClip puts constraints on the maximum number of each cluster being sampled, and therefore avoids overfitting of small clusters.

[Alcoforado et al. (2024)] comprehensively compare different geometry-based diversity sampling techniques such as similarity or distance-based greedy sampling and clustering-based sampling. It proposes three approaches to select subsets $S_b$ for human annotation: 1) reverse semantic search, 2) ordered clustering, and 3) limited lexical similarity. For the reverse semantic search, two datapoints $(x_i, x_j)$ that share the least

semantic similarity are first sampled as $S_b^0$ and then iterative selection of the next most dissimilar element from $S$ is added into $S_b^0$. Its implementation is quite similar to the k-center greedy algorithm (see Alg. 2) except for the initialization of $S_b^0$. For the limited lexical similarity approach, the first sample $x_0$ is chosen randomly for initialization of $S_b^0$. For the remaining $b-1$ quota, each sample $x_i$ is also randomly chosen from $S \backslash S_b$ as long as $sim(x_i, x_{i-1}) \le \tau$, where $sim(\cdot, \cdot)$ here denotes the lexical similarity such as BLEU [Papineni et al. (2002)] and ROUGE scores [Lin (2004)]. The ordered clustering applies a hierarchical and density-based clustering algorithm like HDBSCAN [Campello et al. (2013)] on all samples and sequentially (i.e., from large to small clusters) choose the samples of the lowest membership in each cluster into the subset $S_b$. Experimental results show that the reverse semantic search performs most consistently and competitively, while the limited lexical similarity is sensitive to the hyper-parameter threshold $\tau$. The ordered clustering is not robust across datasets and fails to select high-quality samples.

**Remark** Geometry-based sampling is intuitive and effective in diversity control. Most solutions to optimizing the overall diversity can be reformulated as variants of an iterative, similarity or distance-based, greedy sampling technique. Clustering does play an explanatory role in deciphering the embedding structures, making it easier and preciser to control the proportion of selection.

### 4.4 Bilevel Optimization-based Coreset Sampling

**Overview** The selection of coreset can also be viewede as a bilevel optimization problem [Colson et al. (2007); Zhang (2024); Sinha et al. (2017); Borsos et al. (2020); Killamsetty et al. (2021b;c); Zhang et al. (2022); Borsos et al. (2024); Pan et al. (2024)] that consists of two loops: 1) the outer loop of optimizing the hard masks or soft weights for selecting the subset $S_b$ from $S$; 2) the inner loop of optimizing the model parameters $\theta$ on $S_b$. Without lose of generalizability, the bilevel optimization with the self-supervised language modeling loss can be written as follows:

$$
\begin{aligned}
S_b &= \arg\min_{S_b' \subset S} \sum_{x_i \in S_b', \theta = \theta^*} NLL_i^{A|Q}, \\
&\text{s.t. } \theta^* = \arg\min_\theta \sum_{x_i \in S_b'} NLL_i^{A|Q}.
\end{aligned}
\tag{45}
$$

**Technical Details** The retrieve method proposed by [Killamsetty et al. (2021c)] takes both labeled and unlabeled datasets into consideration, where the self-supervised loss from the unlabeled set (e.g., consistency regularization [Xie et al. (2020); Wang et al. (2021c)] and entropy regularization [Zhao et al. (2020b); Grandvalet & Bengio (2004); Erkan & Altun (2010)]) contributes to the inter-level and outer-level optimization as well. To improve the robustness, Glister [Killamsetty et al. (2021b)] optimizes the outer-level coreset selection on the additionally prepared validation set for the minimized validation loss. [Li et al. (2023e)] further emphasize the role of the validation set in bilevel optimization. It not only computes the loss on the validation set for adversarial training, but also introduces gradient matching [Killamsetty et al. (2021a)] where the gradient of the model on the selected subset $S_b$ should be close to that on the entire $S$.

[Borsos et al. (2024)] reformulate the coreset sampling as a cardinality-constrained bilevel optimization problem. It proposes greedy forward selection and first-order methods that apply to any twice differentiable models. Variants of the solution for acceleration are extended: 1) binary weights, inverse-hessian-vector product approximations, and batch-wise selection; 2) small proxy models for fast estimation; 3) enforced sparsity-inducing penalty in the outer loop.

The ScaleBiO [Pan et al. (2024)] specifically address the data reweighting problem for large-scale LLM instruction tuning. It also prepares an extra validation set $S^{val}$ for the minimization of the outer loop. ScaleBio transforms the bilevel optimization into the single loop framework with an outer-level problem plus a constraint of the inner-level problem. A multiplier $\alpha > 0$ and a proxy $u$ for optimizing the original inner loop (i.e., model weights $\theta$) are introduced into the minimax formulation [Kwon et al. (2023); Lu & Mei (2024)].

In contrast to a fixed budget $b$, [Xia et al. (2024c)] propose a lexicographic bilevel-optimization method [Borsos et al. (2020); Killamsetty et al. (2021b;c)] where the inner loop optimizes model parameters and the outer

loop optimizes data selection. During optimization of the data selection mask, the loss terms are relaxed to allow the size of the final coreset smaller than $b$.

**Remark**  The bilevel optimization methods often involve regularization tricks as a relaxation to the original problem with nested outer-inner loops. Compared with the boolean selection, soft weights-based objective guarantees a higher level of diversity as each sample contributes more or less to the overall optimization.

## 5 Importance-based Selection

This section provides the review of methods on importance measurement and selection. By importance we mean the necessity of adding one instruction-response sample into the training set. Due to the pre-training nature of LLMs, a wide range of materials have been "parameterized" as internal knowledge and therefore several common tasks can be correctly solved without additional fine-tuning. In this case, alignment is not required for easy samples but becomes indispensable for difficult ones. The selected datapoints provide supplementary knowledge to activate the pre-trained LLMs on following complex instructions. Generally, the evaluation function $q(x_i)$ in the context of importance can be alternatively implemented as: 1) the complexity of the datapoint itself $q_C$ that can be perceived by the task definition and the model's uncertainty, 2) the contribution of each datapoint to the overall performance $q_P$ that can be reflected in the losses and errors during learning, and 3) the degree of gradient matching $q_G$ that indicates influential datapoints for effective selection. Consequently, we can unify the importance-based measurements as:

$$q(x_i) = f_i(q_C(x_i), q_P(x_i), q_G(x_i)), \tag{46}$$

where $f_i$ denotes the aggregation function. It is noted that most existing methods simply choose one of the above evaluation implementations and therefore $f_i$ can be designed as an exclusive choice function.

### 5.1 Hand-crafted Indicators

**Overview**  Existing researches on importance measurement of datapoints often stem from two aspects: 1) from the perspective of a datapoint itself, i.e., the difficulty or complexity of each datapoint and the amount of information it provides; 2) from the perspective of the model under development, i.e., the necessity of learning from such a datapoint based on the current performance and confidence (uncertainty), Most hand-crafted indicators are proposed to analyze the text difficulty.

**Technical Details**  The readability indices [Young & Shishido (2023)] can be used to assess both quality (see Sec. §3.1) and difficulty of text samples. Specifically, samples with intricate grammar, advanced vocabulary, and inference dependency are deemed as difficult ones and can be used to evaluate robustness of models across benchmarks of various difficulty levels [Smith & Johnson (2020); Kiela et al. (2021); Ethayarajh et al. (2022); Belinkov & Glass (2019); Nie et al. (2019); Ribeiro et al. (2020)]. For specialized domains such as solving maths problems, the education level (e.g., elementary-level, high school-level, and university-level) determines the difficulty of samples [Patel et al. (2021); Huang et al. (2016); Koncel-Kedziorski et al. (2016)].

One of the pioneering studies on readability scores for difficulty assessment is to compute the percentage of difficult or easy words in one sentence [Klare (1974); Begeny & Greene (2014)]. The words on a pre-defined list are counted as familiar words, and those not listed are unfamiliar, advanced words. Besides, the average number of syllables per word, the number of single-syllable words, and the number of multi-syllable words are also indicative in assessing the text materials [Connatser (1999); Carrell (1987); Zakaluk & Samuels (1988); Dale & Chall (1949)]. Notably, there exist three representative readability metrics: 1) the Dale Chall formula [Chall & Dale (1995)], 2) the flesch reading ease [Flesch (1948)], and 3) the gunning fog index [Gunning (1952)]. Given these metrics, [Saranathan et al.] conduct a thorough analysis on existing NLP datasets $S$ to select the most challenging subsets for efficient evaluation of LLMs. The easiest and hardest samples from the TruthfulQA [Lin et al. (2021)] via these indicators are confirmed positively correlated with the actual complexity. The selection of difficult instruction-response pairs via Eq. 8 allows a wider distribution of performance for the models under investigation. The introduction of difficult datapoints into the subset $S_b$ helps keep the relative rank of different models unchanged compared with that measured on the entire set $S$.

**Remark** The difficulty indices help comprehensively analyze the robustness of models across samples and datasets. In addition, it also presents guidelines of curating and constructing discriminating NLP benchmarks.

## 5.2 Model-based Indicators

**Overview** To avoid potential confusion, the model-based importance indicators discussed in this section are mainly categorized as three kinds: 1) uncertainty-based; 2) reward score-based; and 3) data model-based. Methods that employ training/inference losses, errors (metrics), and gradients are not included despite their involvement of the language model for importance sampling.

**Technical Details** Inspired from uncertainty indicators, [Siddhant & Lipton (2018); Kung et al. (2023); Nieth et al. (2024)] propose the prompt uncertainty which measures the disagreement between model responses on different perturbed versions of the same instruction:

$$
U_i^{\text{prompt}} = -\frac{1}{K} \sum_{k=1}^{K} \sum_{j=t}^{|x_i|} |P(x_{i(j)}|x_{i(<j)}; \theta) -
$$
$$
P(x_{i(j)}|\tilde{x}_{i(<j)}^k; \theta)|,
\tag{47}
$$

where $K$ denotes the number of perturbations and $\tilde{x}_i^k$ is the $k$-th perturbed prompt. Note that only the instruction part $x_{i(<t)}$ is perturbed and sent to the model for the following likelihood measurement on the original response $x_{i(j)}, j = t, t+1, ..., |x_i|$. Samples with high prompt uncertainty should be chosen for fine-tuning since the model does not perform consistently on such instructions.

[Jiang et al. (2023b)] target at the over-confidence problem of LLMs after instruction tuning [Kadavath et al. (2022)], and propose the CAPE to calibrate the uncertainty with augmented prompt ensembles. They first transform all discriminative and generative tasks into multiple-choice problems, and use the LLM's predicted probabilities over answer choices (e.g., A, B, C) for uncertainty estimation. Then, prompt augmentations are performed via paraphrasing of the template, permutation of the in-context examples and the answer choices. Multiple predictions over the answer choices are collected with the augmented prompts as inputs, which helps calibrate the uncertainty in an ensemble manner. Such calibrated uncertainty tells if an instruction-tuned LLM simply memorizes the response to a given prompt rather than truly understanding the instruction. Therefore, it can be used to precisely choose important datapoints with high uncertainty.

Apart from the uncertainty, the reward model can also be used beyond quality scorer. Since most of the knowledge and capabilities are acquired during pre-training [Zhou et al. (2024a)], the instruction tuning datasets are aimed at aligning the behavior of models with human preference and expectations. Therefore, for any given instruction $x_i$, if the generated response is of high quality, then the necessity of fine-tuning on this instruction is low. Accordingly, $x_i$ is deemed as "unimportant" and will not be chosen into the subset. In that case, the language model parameterized as $\theta$ is first prompted with $x_{i(<t)}$ to generate the response $\hat{x}_{i(\geq t)}^\theta$. Then, a reward model parameterized as $\phi$ acts as a necessity evaluation model:
$$
\hat{R}_i = r_\phi(x_{i(<t)}, \hat{x}_{i(\geq t)}^\theta).
\tag{48}
$$

Samples whose necessity score $\hat{R}_i$ below a pre-determined threshold are selected via Eq. 7, implying that the model $\theta$ does not own the capabilities to handle $x_i$ and requires fine-tuning.

Another series of model-based importance estimation methods are based on datamodels [Ilyas et al. (2022); Park et al. (2023); Jain et al. (2023); Kang et al. (2024); Chhabra et al. (2024); Saunshi et al. (2022); Ye et al. (2024)], where the contribution of each datapoint to the model's behavior is estimated. The datamodels can be implemented in any machine learning model which targets at predicting the influence of each datapoint on the performance of the trained model [Koh & Liang (2017); Jain et al. (2022); Liu et al. (2024b); Picard et al. (2024); Bae et al. (2024); Covert et al. (2024)].

[Engstrom et al. (2024)] propose to use datamodels to select subsets that maximize the overall performance. Specifically, it chooses the subset $S_b \subset S, S = \{x_1, x_2, ..., x_{|S|}\}$ by estimating the loss of the model trained on it. Out of simplicity, the datamodel $\tau_{\theta_x}$ can be implemented as a linear model and it learns to approximate the actual loss via the TARK estimator [Park et al. (2023)]:

$$\theta_{x_j} = \arg\min_{\theta} \hat{\mathbb{E}}^{(m)}_{S_i \sim S_b \subset S}[L_{reg}(\tau_\theta(\mathbb{1}_{S_i})), \mathcal{L}_{x_j}(S_i)],$$

$$\mathbb{1}_{S_b} \in \{0,1\}^{|S|}, \quad (\mathbb{1}_{S_b})_i = \begin{cases} 1, & \text{if } x_i \in S_b, \\ 0, & \text{otherwise.} \end{cases}, \quad \tau_{\theta_x}(\mathbb{1}_{S_b}) = \theta_x^T \mathbb{1}_{S_b}, \tag{49}$$

where $\mathcal{L}_{x_j}(S_i)$ denotes the loss of the model (trained on $S_i$) on the sample $x_j$. The $\hat{\mathbb{E}}^{(m)}$ is a $m$-sample empirical expectation and $L_{reg}(\cdot,\cdot)$ is a regression loss function (e.g., mean squared error). Intuitively, what the datamodel $\tau_{\theta_x}$ does is to approximate the real loss $\mathcal{L}_{x_j}(S_i)$ under various compositions of subsets $S_i \sim S_b$. Given any subset $S'_b$, the averaged loss approximated by the datamodel on all $x_j \in S_{eval}$ is calculated on the evaluation set $S_{eval}$ and minimized to find the optimal $S_b, |S_b| = b$:

$$S_b = \arg\min_{S'_b \subset S} \hat{\mathcal{L}}_{S_{eval}}(S'_b),$$

$$\begin{aligned} \hat{\mathcal{L}}_{S_{eval}}(S'_b) &= \hat{\mathbb{E}}^{(n)}_{x_j \sim S_{eval}}[\tau_{\theta_{x_j}}(\mathbb{1}_{S'_b})] \\ &= \frac{1}{|S_{eval}|} \sum_{x_j \in S_{eval}} \theta_{x_j}^T \mathbb{1}_{S_b^*} = \mathbb{1}_{S_b^*}^T (\frac{1}{|S_{eval}|} \sum_{x_j \in S_{eval}} \theta_{x_j}). \end{aligned} \tag{50}$$

The importance of $x_i \in S$ is therefore measured by $\frac{1}{|S_{eval}|} \sum_{x_j \in S_{eval}} \theta_{x_j}$ and its smallest $b$ elements are chosen for the minimum loss $\hat{\mathcal{L}}_{S_{eval}}$.

[Liu et al. (2024b)] also propose a simulation-based [Guu et al. (2023)] linear datamodel that correlates the training samples with the validation or test set loss. A featured simulator, namely GPTfluence, models the training dynamics (e.g., loss, BLEU and ROUGE scores) across time via an $n$-th order Markov process. It extracts representations $g(x_i), x_i \in S$ from BERT or GPT, and generates both multiplicative and additive factors to reflect the influence of any training example on the testing set. The testing performance $\phi_t$ at any time $t$ is affected by: 1) its performance at preceding $n$ times and 2) the current training batch $c_t$:

$$\phi_t(x_k) = \sum_{j=1}^{n} \alpha_j(c_t)\phi_{t-j}(x_k) + \beta(c_t), \forall x_k \in S_{eval},$$

$$\alpha_j(c_t) = \sum_{i=1}^{|c_t|} A_{i,j}, \quad \beta(c_t) = \sum_{i=1}^{|c_t|} B_i, \forall x_i \in c_t \subset S, \tag{51}$$

$$A_{ij} = \langle \mathbf{W}_{(j)}^T g(x_i)_j, \mathbf{U}_{(j)}^T g(x_k) \rangle_F, \quad B_i = \langle \mathbf{W'}^T g(x_i)_j, \mathbf{U'} g(x_k) \rangle_F,$$

where $\mathbf{W}_{(j)}^T, \mathbf{U}_{(j)}^T, \mathbf{W'}, \mathbf{U'}$ are learnable weights which are optimized by minimizing $\sum_{t=1}^{T}(y_t - \phi_t(x_k))^2$ with $y_t$ being the ground-truth metric score monitored during training at step $t$. The $\langle \cdot, \cdot \rangle_F$ denotes the Frobenius inner product. Samples that reduce the evaluation loss the most are selected as influential data.

Instead of performing off-line selection, [Yu et al. (2024)] propose MATES where a small datamodel continuously selects the most effective subset for the current training of the LLM. The datamodel, like a partner, is updated alternatively to adapt to the constantly changing preferences of the model under development.

Unlike previous datamodels that predict the influence of datapoints via the testing performance of the model, [Xie et al. (2023)] propose the DSIR with importance scores estimated by the distributional resemblance. It simply assumes that training samples that resemble the evaluation set are important, and these datapoints should be selected with higher probability. Given the hashed n-grams features $h(x_i) \in \mathbb{N}^m$ of $x_i$, its importance score $w_i$ is calculated as:

$$w_i = \frac{\hat{w}_i}{\sum_{i=1}^{|S|} \hat{w}_i}, \quad \hat{w}_i = \frac{\hat{p}_{feat}(h(x_i))}{\hat{q}_{feat}(h(x_i))},$$

$$\hat{p}_{feat}(h(x_i)) = \prod_{j=1}^{m} \gamma_j^{h(x_i)_j}, \quad \hat{q}_{feat}(h(x_i)) = \prod_{j=1}^{m} \beta_j^{h(x_i)_j}, \tag{52}$$

$$\hat{\gamma} = \frac{1}{\sum_{x_i \in S_{eval}} \mathbf{1}^T h(x_i)} \sum_{x_j \in S_{eval}} h(x_j), \quad \hat{\beta} = \frac{1}{\sum_{x_i \in S} \mathbf{1}^T h(x_i)} \sum_{x_j \in S} h(x_j),$$

where $S$ and $S_{eval}$ respectively denote the training set and the evaluation set. Given the budget $b$, the subset $S_b$ is obtained by importance-weighted sampling without replacement $b$ times. [Zhang et al. (2023c)] also proposes to use a independent-cascade diffusion model [Li et al. (2018); Du et al. (2014)] to mimic the information diffusion process upon a directed graph on embeddings of datapoints. The most influential datapoint are selected for annotation and serve as in-context learning examples for LLMs.

**Remark**   Compared with uncertainty and reward score, datamodel-based importance indicators are more correlated with the downstream performance since the task-specific evaluation set is introduced to provide feedback on the selection scheme.

### 5.3   Loss and Error-based Coreset Sampling

**Overview**   During training, samples that contribute more to the total  loss or cause worse performance are considered more important. In the light of this statement, the influence of each datapoint can also be measured via the losses and errors for coreset sampling. Compared with the datamodel-based measurement that estimates individual importance with a specifically designed datamodel, loss and error-based  measurement is performed with the same LLM under development.

**Technical Details**   One kind of methods that record the errors of each sample during training to estimate importance is forgetting score or forgetting event [Toneva et al. (2018)]. It counts how many times the forgetting happens with the iteration of training step $t$. For any given sample $x_i$ in a batch $B$ ($x_i \in B \subset S$), if the previous accuracy $acc_i^{t-1}$ surpasses the current accuracy $acc_i^t$ ($acc_i^t > acc_i^{t+1}$), then the example $x_i$ undergoes a forgetting event. Conversely, a learning event occurs if $acc_i^t < acc_i^{t+1}$. The number of forgetting events implies whether the sample is difficult and indispensable for training. An example $x_i$ is defined as unforgettable if it satisfies:

$$Unforget_i = \begin{cases} 1, & \exists t^* < \infty, \text{ s.t. } acc_i^t < acc_i^{t+1} \\ & \text{and } \forall k \geq t^*, acc_i^k > acc_i^{k-1}, \\ 0, & \text{otherwise.} \end{cases} \tag{53}$$

The easy samples with $Unforget_i = 1$ can be simply discarded and the important subset $S_b = \{x_i | Unforget_i = 0, x_i \in S\}$ is selected for training. Recent studies on both pre-training and instruction tuning have investigated the effectiveness of using the forgetting score for efficient data pruning [Sorscher et al. (2022); Paul et al. (2021); Zhang et al. (2023a); Jin & Ren (2024a); Maini et al. (2022)].

In contrast to the term "forgetting", researchers introduce the concept "memorization" [Feldman (2020); Tirumala et al. (2022); Antoniades et al. (2024)] for analysis on the generalization of deep models [Zhang et al. (2021)]. The memorization of training samples is necessary for reducing close-to-optimal generalization error especially when a long-tailed disttribution is observed for the training set [Feldman (2020)]. Specifically, the amount of label memorization on the instruction-response pair $(x_{i(<t)}, x_{i(\geq t)})$ is defined as follows:

$$Memo_i = \frac{1}{|x_i| - t} \sum_{j=t}^{|x_i|} (P(x_{i(j)} | x_{i(<j)}; \theta^S) - $$
$$P(x_{i(j)} | x_{i(<j)}; \theta^{S \backslash x_i})), \tag{54}$$

where $\theta^S$ and $\theta^{S \backslash x_i}$ respectively refer to the language model parameters optimized with the entire set with and without $x_i$. Accordingly, the influence of the sample $x_i$ on other samples $x_k, x_k \neq x_i$ can be defined as [Feldman & Zhang (2020)]:

$$Infl_{ik} = \frac{1}{|x_k| - t} \sum_{j=t}^{|x_k|} (P(x_{k(j)} | x_{k(<j)}; \theta^S) - $$
$$P(x_{k(j)} | x_{k(<j)}; \theta^{S \backslash x_i})), \tag{55}$$

where $x_{k(<t)}$ and $x_{k(\geq t)}$) respectively denote the instruction and response part of $x_k$. In practice, the memorization and influence scores are approximated via batch-wise sampling where $N$ batches $B_1, B_2, ..., B_N$ are sampled from $S$ with $|B_i| = n$. For each batch $B_i$, a language model parameterized as $\theta^{B_i}$ is trained to compute the memorization and influence scores of each sample $x_i$. It is noted that some batches contain $x_i$ and the others do not. Therefore, the two probability terms in Eqs. 54 and 55 are respectively averaged over multiple probability outputs of the models trained on batches with and without $x_i$. [Sorscher et al. (2022)] confirm that memorization scores (Eq. 54) demonstrate stronger performance on pruning the dataset into a significantly smaller subset $S_b$ than random sampling, EL2N (Eq. 13), and influence scores (Eq. 55). [Suzuki et al. (2023)] and [Schoch et al. (2023)] also follow [Feldman & Zhang (2020)] to select the high-quality influential subset for LLM training.

Furthermore, [Chen et al. (2024b)] use the evaluation loss to check whether the current task requires certain skills or capabilities that can be obtained by learning from the prerequisite tasks. For each task, it selects the skill-dependent datapoints that reduce evaluation loss. [Mishra & Sachdeva (2020)] propose a rather simple method that adopts a proxy model (e.g., logistic regression and SVM) to train on the randomly selected subset $S_b$ and evaluate on the remaining set $S \backslash S_b$. Such process iterates over multiple times to ensure that each sample is at least validated once. The probability of each sample being correctly predicted is used as importance measurement. Likewise, [Lin et al. (2022)] also quantify the average marginal effect (AME) as influence of $x_i$. It can be viewed as a variant of shapley value [Jia et al. (2019); Ghorbani & Zou (2019); Schoch et al. (2023); Kwon & Zou (2021)]. Different subsets are randomly sampled to train multiple submodels and each submodel is evaluated for jointly estimating the AME via LASSO regression [Lecué & Mendelson (2018)].

**Remark** The loss and error-based selection methods are intuitive and effective to select the datapoints with high difficulty and influence. To accelerate the computation of marginal effect (gain) of each datapoint, iterative approximations can be adopted with small proxy models.

### 5.4 Gradient-based Coreset Sampling

**Overview** Since gradients directly affect the optimization of language models, two kinds of intuitive methods for data selection are presented: 1) gradient matching [Zhao et al. (2020a); Killamsetty et al. (2021a); Jiang et al. (2023d); Zhao & Bilen (2023); Du et al. (2024); Balles et al. (2022); Zhang et al. (2024a)], i.e., the gradients of the entire set $S$ being approximated by the weighted gradients of the subset $S_b$, and 2) gradient-based influence [Pruthi et al. (2020); Brophy et al. (2023); Koh & Liang (2017); Basu et al. (2020); Picard et al. (2024); Alaa & Van Der Schaar (2020)], i.e., the influence of each sample $x_i$ on a testing datapoint $x_t$ being measured by upweighted gradient multiplication. Specifically, the gradient matching aims to minimize the difference below:

$$
\theta^*, S_b^* = \arg\min_{\theta, S} \ d(\frac{1}{|S|} \sum_{x_i \in S} \nabla_\theta NLL_i^{A|Q},
$$
$$
\frac{1}{\sum_i w_i} \sum_{x_i \in S_b} w_i \nabla_\theta NLL_i^{A|Q}), \ S_b \in S, w_i > 0,
$$
(56)

where $d(\cdot, \cdot)$ denotes the distance measurement and $w_i$ is the weight for the gradient of $x_i$.

The gradient-based influence methods, on the other hand, aim at selecting the most influential datapoints in terms of the variation of model parameters $\theta$. Given the optimal parameters $\theta^*$, the updated parameters $\theta^\epsilon_{\{x_i\}}$ by up-weighting the loss of $x_i$ with $\epsilon$ can be derived as the first-order Taylor series expansion as follows:

$$
\theta^\epsilon_{x_i} = \arg\min_\theta \frac{1}{|S|} \sum_{x_j \in S} NLL_j^{A|Q} + \epsilon NLL_i^{A|Q},
$$
$$
\theta^\epsilon_{x_i} \approx \theta^* - \epsilon H_{\theta^*}^{-1} \nabla_\theta NLL_i^{A|Q},
$$
(57)

where $H_{\theta^*}$ represents the Hessian with respect to the model parameters $\theta^*$. Accordingly, the influence function of a sample $x_i$ on the model parameters and its effect on the performance of a particular sample $x_j$ can be

respectively denoted as:

$$
\begin{aligned}
InflF_i &= \frac{\mathrm{d}\theta^\epsilon_{x_i}}{\mathrm{d}\epsilon}|_{\epsilon=0} = -H^{-1}_{\theta^*}\nabla_\theta NLL^{A|Q}_i, \\
InflF_{ij} &= -\nabla_\theta NLL^{A|Q^T}_j H^{-1}_{\theta^*}\nabla_\theta NLL^{A|Q}_i.
\end{aligned}
\tag{58}
$$

The importance indicator $InflF_{ij}$ approximately measures the change of the loss on $x_j$ when $x_i$ is removed from the training set. To expedite the computation of Hessian matrix for large models, a combination of Hessian-vector product and optimization techniques are developed [Pearlmutter (1994); Nilsen et al. (2019); Mathieu & LeCun (2014); Agarwal et al. (2016); Shewchuk et al. (1994)].

Another kind of influence score is defined as the expected gradient norm (GraNd score) [Paul et al. (2021); Kirsch (2023); Böther et al. (2023)], where the GraNd score controls the contribution of a training sample to the change of the loss.

$$
GraNd_i = \mathbb{E}_\theta \|\nabla_\theta NLL^{A|Q}_i\|_2
\tag{59}
$$

Experiments [Paul et al. (2021)] suggest that the GraNd score (Eq. 59) can be well approximated by EL2N score (Eq. 13) for efficient data pruning.

**Technical Details**   [Xia et al. (2024a)] propose to find the most influential training data that resemble the testing set the most via low-rank gradient similarity search. [Tan et al. (2024a)] introduce the moving-one-sample-out (MoSo) by pinpointing the least informative samples via gradient-based influence assessment. To avoid the costly retraining procedure by iteratively moving one sample out, a gradient-based approximator is proposed to select samples whose gradients are consistently aligned with the average gradients of the entire training set.

For the detailed definition of distance measure of Eq. 56, [Everaert & Potts (2023)] exploit the KL-divergence to measure the difference between the selected subset and the testing set. Note that here the objective is to approach the distribution of the testing set rather than the entire training set. [Killamsetty et al. (2021a)] speed up the gradient matching between the selected dataset and the validation set via an orthogonal matching pursuit algorithm. [Lin et al. (2024)] apply gradient-based influence scores on recommendation datasets for effective LLM instruction tuning. [Schioppa et al. (2021)] choose a different way [Arnoldi (1951)] to accelerate the computation of the inverse Hessian matrix in Eq. 57 and successfully scales up the influence scoring for LLMs with several hundreds of millions of parameters. [Grosse et al. (2023)] use the influence functions to study the generalization properties of LLMs To scale up influence functions for LLMs up to 52 billions, an approximation technique via Eigenvalue-corrected Kronecker-Factored Approximate Curvature (EK-FAC) [George et al. (2021)] to efficiently find the most influential samples to the pre-trained LLMs over maths and programming abilities, cross-lingual generalization, and role-playing behavior. [Zhao et al. (2021)] condense the datasets into small informative synthetic samples where the gradients of the model on the synthetic data are matching those on the real data of the entire training set.

**Remark**   The gradient-based coreset sampling techniques are highly dependent on the LLMs under development, where the gradients describe the model's inherent knowledge and uncertainty about each datapoint. Despite the precision of gradient-based selection methods, it is noted that approximation is unavoidable for application on LLMs. The efficiency and accuracy of various approximation techniques should be considered.

## 6   Results and Discussions

In this section, we classify different methods according to their respective  emphases, and then demonstrate their effectiveness in the selection of "high-standard" datapoints for instruction tuning. First, we provide explanations on the classification and selection of recent instruction tuning methods. Then, we summarize the representative methods with their data statistics in Tab. 1. Finally, we provide the detailed experimental results of each method respectively under our structures of quality, diversity, and importance.

**Criteria on Classification and Selection of Existing Methods**   For the classification of existing methods into quality-based, diversity-based, and importance-based categories, we adhere to the following

Table 1: Statistics of datasets in existing representative data assessment and selection methods.

| Methods | Quality | Diversity | Importance | Training Set | Training Set Size |
|---|---|---|---|---|---|
| IFD [Li et al. (2023a)] | ✓ | ✗ | ✗ | Alpaca
WizardLM | 52K
70K |
| LIFT [Xu et al. (2023b)] | ✓ | ✓ | ✗ | Open-Platypus
CodeAlpaca | 25K
20K |
| DQ [Zhou et al. (2023)] | ✗ | ✓ | ✗ | Alpaca | 52K |
| PPL [Ankner et al. (2024)] | ✓ | ✗ | ✗ | The Pile
Dolma | NA
NA |
| InstructionMining [Cao et al. (2023)] | ✓ | ✗ | ✗ | OpenOrca
Dolly | 50K
15K |
| FL [Bhatt et al. (2024)] | ✓ | ✓ | ✗ | FLAN v2 | 99K |
| Alpagasus [Chen et al. (2023b)] | ✓ | ✗ | ✗ | Alpaca | 52K |
| BSDetector [Chen & Mueller (2024)] | ✓ | ✗ | ✗ | SQuAD-N
Emails-N
DROP-N | NA
NA
NA |
| DEITA [Liu et al. (2023b)] | ✓ | ✓ | ✗ | Mixed(ShareGPT+UltraChat+WizardLM) | 206K |
| AutoDS [Zhang et al. (2024c)] | ✓ | ✗ | ✗ | OpenWebMath | NA |
| Qurator [Wettig et al. (2024)] | ✓ | ✗ | ✗ | QuRatedPajama | 260B tkn |
| ClusterClip [Shao et al. (2024)] | ✗ | ✓ | ✗ | OpenOrca
Proof-Pile-2 | 4.2M
2.7M |
| QDIT [Bukharin & Zhao (2023)] | ✓ | ✓ | ✗ | UltraChat
LMSYS
Alpaca
Mixed (Alpaca+OIG+Dolly)
Dolly | 1.3M
1M
52K
270K
15K |
| DsDm [Engstrom et al. (2024)] | ✗ | ✗ | ✓ | C4 | NA |
| MATES [Yu et al. (2024)] | ✗ | ✗ | ✓ | C4 | NA |
| DSIR [Xie et al. (2023)] | ✗ | ✗ | ✓ | The Pile | 1.6B |
| Skill-it [Chen et al. (2024b)] | ✗ | ✗ | ✓ | RedPajama | 1.2T tokens |
| LESS [Xia et al. (2024a)] | ✗ | ✗ | ✓ | Mixed(FLAN v2+Dolly+OpenAssistant+COT) | 270K |

evidence: 1) the data characteristics emphasized in the motivation of their method development, and 2) the optimization objectives organized in this survey that are closest to those in their pipelines. For instance, the concept of diversity is stressed in DEITA [Liu et al. (2023b)] where a similarity-based filtering step is set to remove highly duplicated datapoints. Therefore, it falls under the category of diversity-based selection methods. DSIR [Xie et al. (2023)], on the other hand, directly estimates the importance of datapoints by matching the distribution between the selected subset and the target evaluation set. It is noted that many existing methods design compound, multi-facet selection criteria that strike a balance between quality, diversity, and importance. In this case, we choose the most representative methods under each category to highlight their corresponding facets without misinterpreting their mechanisms. Furthermore, we select the most recent methods that exploit open-sourced LLMs for verifying the effectiveness of their proposed data selection techniques. Such accessibility to public LLMs avoids the privacy issues and inference costs brought by requesting API services from close-sourced proprietary models.

**Quality** The quality of data directly impacts the effectiveness of model training. Quality control measures include data scoring, quality assessment, and more. In Tab. 2, we have summarized the results of different methods focusing on data quality. In the table, we list the data used by different methods and the proportion/size of the data selected. It can be seen that the method of selecting data based on quality can match the results of training with full data even under the data-poor regime. They are also superior to the results of randomly selecting a subset from the original dataset. In the table, WK stands for World Knowledge, CR stands for Commonsense Reasoning, LU stands for Language Understanding, SPS stands for Symbolic Problem Solving, and RC stands for Reading Comprehension.

**Diversity** Data engineering enhances the generalization ability of models by improving the diversity of datasets. Such improved diversity may be implemented via encompassing data from different sources, with varying features, and of distinct distributions. Researches indicate that it is insufficient to merely select datasets that are similar to the target data from the downstream tasks. Tab. 3 demonstrates the importance of diversity in data selection. Compared with random selection and uniform selection, the scheme of selecting

Table 2: Experimental results of quality-based selection methods are directly cited from their papers. BT denotes billions of tokens. WK, CR, LU, SPS, and RC respectively stand for compound datasets of World Knowledge, Commonsense Reasoning, Language Understanding, Symbolic Problem Solving, and Reading Comprehension.

**IFD [Li et al. (2023a)]**

| Training Set | Model | Ratio/Size | ARC | HellaSwag | MMLU | TruthfulQA | AlpacaEval |
|---|---|---|---|---|---|---|---|
| Alpaca | LLaMA 7B | Full | 0.427 | 0.769 | 0.417 | 0.396 | 0.265 |
| Alpaca | LLaMA 7B | 5% | 0.539 | 0.795 | 0.365 | 0.383 | 0.347 |
| WizardLM | LLaMA 7B | Full | 0.531 | 0.774 | 0.378 | 0.429 | 0.620 |
| WizardLM | LLaMA 7B | 10% | 0.529 | 0.790 | 0.331 | 0.414 | 0.614 |
| Alpaca | LLaMA2 7B | Full | 0.544 | 0.787 | 0.470 | 0.410 | 0.278 |
| Alpaca | LLaMA2 7B | 5% | 0.558 | 0.579 | 0.804 | 0.442 | 0.368 |
| Alpaca | LLaMA2 7B | 10% | 0.580 | 0.804 | 0.466 | 0.402 | NA |
| Alpaca | LLaMA2 7B | 15% | 0.564 | 0.574 | 0.807 | 0.464 | NA |
| WizardLM | LLaMA2 7B | Full | 0.576 | 0.820 | 0.541 | 0.415 | 0.350 |
| WizardLM | LLaMA2 7B | 5% | 0.624 | 0.840 | 0.557 | 0.428 | 0.468 |
| WizardLM | LLaMA2 7B | 10% | 0.630 | 0.839 | 0.553 | 0.419 | NA |
| WizardLM | LLaMA2 7B | 15% | 0.624 | 0.835 | 0.556 | 0.434 | NA |

**LIFT [Xu et al. (2023b)]**

| Training Set | Model | Ratio/Size | ARC | HellaSwag | MMLU | TruthfulQA |
|---|---|---|---|---|---|---|
| Open-Platypus | Mistral 7B | Random 15K | 0.607 | 0.820 | 0.625 | 0.438 |
| Open-Platypus | Mistral 7B | LIFT 15K | 0.643 | 0.844 | 0.645 | 0.490 |

| Training Set | Model | Ratio/Size | HumanEval | MBPP |
|---|---|---|---|---|
| Code-Alpaca | StarCoder 15B | Random 10K | 0.381 | 0.431 |
| Code-Alpaca | StarCoder 15B | LIFT 10K | 0.550 | 0.495 |

**PPL [Ankner et al. (2024)]**

| Training Set | Model | Ratio/Size | WK | CR | LU | SPS | RC |
|---|---|---|---|---|---|---|---|
| The Pile | MPT 1B | Full | 0.155 | 0.103 | 0.281 | 0.035 | 0.112 |
| The Pile | MPT 1B | Low 50% | 0.111 | 0.058 | 0.187 | 0.035 | 0.087 |
| The Pile | MPT 1B | Mid 50% | 0.161 | 0.090 | 0.281 | 0.034 | 0.109 |
| The Pile | MPT 1B | High 50% | 0.182 | 0.128 | 0.332 | 0.034 | 0.106 |
| Dolma | MPT 1B | Full | 0.165 | 0.123 | 0.289 | 0.036 | 0.080 |
| Dolma | MPT 1B | Low 50% | 0.161 | 0.101 | 0.273 | 0.345 | 0.079 |
| Dolma | MPT 1B | Mid 50% | 0.180 | 0.130 | 0.319 | 0.034 | 0.104 |
| Dolma | MPT 1B | High 50% | 0.167 | 0.131 | 0.311 | 0.032 | 0.086 |

**InstructionMining [Cao et al. (2023)]**

| Training Set | Model | Ratio/Size | ARC | HellaSwag | MMLU | TruthfulQA |
|---|---|---|---|---|---|---|
| OpenOrca & Dolly | LLaMA2 7B | IM 10K | 0.567 | 0.798 | 0.499 | 0.483 |
| OpenOrca & Dolly | LLaMA2 7B | IM 40K | 0.544 | 0.801 | 0.526 | 0.498 |
| OpenOrca & Dolly | LLaMA2 7B | Random 10K | 0.548 | 0.796 | 0.490 | 0.516 |
| OpenOrca & Dolly | LLaMA2 7B | Random 40K | 0.548 | 0.799 | 0.512 | 0.500 |

**FL [Bhatt et al. (2024)]**

| Training Set | Model | Ratio/Size | MMLU | BBH |
|---|---|---|---|---|
| FLAN v2 | LLaMA2 7B | Random 20K | 0.443 | 0.390 |
| FLAN v2 | LLaMA2 7B | FL 20K | 0.451 | 0.383 |
| FLAN v2 | LLaMA2 7B | Random 30K | 0.449 | 0.394 |
| FLAN v2 | LLaMA2 7B | FL 30K | 0.471 | 0.411 |
| FLAN v2 | LLaMA2 7B | Random 45K | 0.460 | 0.394 |
| FLAN v2 | LLaMA2 7B | FL 45K | 0.476 | 0.413 |

**Alpagasus [Chen et al. (2023b)]**

| Training Set | Model | Ratio/Size | BBH | DROP | HumanEval | MMLU |
|---|---|---|---|---|---|---|
| Alpaca | LLaMA2 7B | Random 9K | 0.319 | 0.259 | 0.116 | 0.369 |
| Alpaca | LLaMA2 7B | Full 52K | 0.330 | 0.259 | 0.117 | 0.409 |
| Alpaca | LLaMA2 7B | Alpagasus 9K | 0.338 | 0.260 | 0.122 | 0.388 |
| Alpaca | LLaMA2 13B | Random 9K | 0.386 | 0.334 | 0.152 | 0.450 |
| Alpaca | LLaMA2 13B | Full 52k | 0.387 | 0.338 | 0.157 | 0.479 |
| Alpaca | LLaMA2 13B | Alpagasus 9K | 0.389 | 0.344 | 0.159 | 0.461 |

**BSDetector [Chen & Mueller (2024)]**

| Training Set | Model | Ratio/Size | SQuQA-N | Emails-N | DROP-N |
|---|---|---|---|---|---|
| SQuAD-N | LLaMA2 7B Chat | Full | 0.499 | NA | NA |
| SQuAD-N | LLaMA2 7B Chat | Auto-filter | 0.599 | NA | NA |
| SQuAD-N | LLaMA2 7B Chat | Auto-correct | 0.714 | NA | NA |
| Emails-N | LLaMA2 7B Chat | Full | NA | 0.507 | NA |
| Emails-N | LLaMA2 7B Chat | Auto-filter | NA | 0.497 | NA |
| Emails-N | LLaMA2 7B Chat | Auto-correct | NA | 0.523 | NA |
| DROP-N | LLaMA2 7B Chat | Full | NA | NA | 0.447 |
| DROP-N | LLaMA2 7B Chat | Auto-filter | NA | NA | 0.474 |
| DROP-N | LLaMA2 7B Chat | Auto-correct | NA | NA | 0.505 |

**AutoDS [Zhang et al. (2024c)]** — Open-WebMath, Mistral 7B

| Ratio/Size | MATH | GSM8K | BBH | ARC-E | ARC-C |
|---|---|---|---|---|---|
| Random 2.5BT | 0.143 | 0.441 | 0.565 | 0.842 | 0.567 |
| AutoDS 2.5BT | 0.161 | 0.454 | 0.586 | 0.842 | 0.552 |

| Ratio/Size | LogiQ | BoolQ | NQ | MMLU | HellaSwag |
|---|---|---|---|---|---|
| Random 2.5BT | 0.310 | 0.838 | 0.292 | 0.522 | 0.622 |
| AutoDS 2.5BT | 0.310 | 0.831 | 0.291 | 0.523 | 0.627 |

| Ratio/Size | PIQA | Winogrande | SciQ |
|---|---|---|---|
| Random 2.5BT | 0.822 | 0.802 | 0.972 |
| AutoDS 2.5BT | 0.822 | 0.800 | 0.968 |

**QuRator [Wettig et al. (2024)]**

| Training Set | Model | Ratio/Size | RC | CR | WK |
|---|---|---|---|---|---|
| QuRated-Pajama | Sheared LLaMA 1.3B | Random 30BT | 0.509 | 0.55 | 0.149 |
| QuRated-Pajama | Sheared LLaMA 1.3B | Qurator 30BT | 0.521 | 0.555 | 0.152 |

Table 3: Experimental results of diversity-based selection methods are directly cited from their papers.

| Methods | Training Set | Model | Ratio/Size | Reported Results on Testing Sets | | | |
|---|---|---|---|---|---|---|---|
| | | | | ARC | HellaSwag | MMLU | TruthfulQA |
| DEITA [Liu et al. (2023b)] | Mixed | LLaMA-13B | Random 10K | 0.558 | 0.800 | 0.474 | 0.574 |
| | | | DEITA 10K | 0.595 | 0.820 | 0.606 | 0.550 |
| | | LLaMA2-13B | Random 10K | 0.615 | 0.837 | 0.552 | 0.448 |
| | | | DEITA 10K | 0.589 | 0.821 | 0.553 | 0.546 |
| | | Mistral-7B | Random 10K | 0.554 | 0.792 | 0.587 | 0.536 |
| | | | DEITA 6K | 0.578 | 0.803 | 0.619 | 0.598 |
| ClusterClip [Shao et al. (2024)] | OpenOrca | Mistral-7B | | SuperGLUE | GSM8k | OBQA | MT-Bench |
| | | | Random 5B tokens | 0.621 | 0.615 | 0.798 | 6.600 |
| | | | Uniform 5B tokens | 0.630 | 0.588 | 0.782 | 6.750 |
| | | | ClusterClip | 0.643 | 0.587 | 0.814 | 6.900 |
| | Proof-Pile-2 | LLaMA2-7B | | MATH | GSM8K | MMLU | BBH |
| | | | Random 20B tkn | 0.065 | 0.256 | 0.488 | 0.418 |
| | | | Uniform 20B tkn | 0.076 | 0.260 | 0.500 | 0.429 |
| | | | ClusterClip | 0.079 | 0.248 | 0.511 | 0.428 |
| QDIT [Bukharin & Zhao (2023)] | | LLaMA-7B | | MMLU | BBH | ARC | |
| | UltraChat | | Random 10K | 0.321 | 0.332 | 0.583 | |
| | | | QDIT 10K | 0.361 | 0.321 | 0.607 | |
| | LMSYS | | Random 10K | 0.331 | 0.326 | 0.602 | |
| | | | QDIT 10K | 0.373 | 0.325 | 0.614 | |
| | Alpaca | | Random 3K | 0.362 | 0.303 | 0.617 | |
| | | | QDIT 3K | 0.355 | 0.304 | 0.620 | |
| | Mixed | | Random 10K | 0.329 | 0.309 | 0.583 | |
| | | | QDIT 10K | 0.343 | 0.312 | 0.607 | |
| | Dolly | | Random 1K | 0.281 | 0.273 | 0.594 | |
| | | | QDIT 1K | 0.338 | 0.303 | 0.598 | |
| | | | | DROP | LAMBADA | SciQ | |
| | UltraChat | | Random 10K | 0.262 | 0.698 | 0.854 | |
| | | | QDIT 10K | 0.267 | 0.698 | 0.868 | |
| | LMSYS | | Random 10K | 0.251 | 0.685 | 0.867 | |
| | | | QDIT 10K | 0.264 | 0.693 | 0.850 | |
| | Alpaca | | Random 3K | 0.263 | 0.716 | 0.870 | |
| | | | QDIT 3K | 0.270 | 0.697 | 0.841 | |
| | Mixed | | Random 10K | 0.203 | 0.681 | 0.841 | |
| | | | QDIT 10K | 0.260 | 0.697 | 0.898 | |
| | Dolly | | Random 1K | 0.173 | 0.717 | 0.807 | |
| | | | QDIT 1K | 0.226 | 0.723 | 0.806 | |
| DQ [Zhou et al. (2023)] | Alpaca | LLaMA-7B | | BBH | DROP | MMLU | HumanEval |
| | | | Full | 0.329 | 0.263 | 0.416 | 0.100 |
| | | | 20% | 0.327 | 0.267 | 0.398 | 0.092 |
| | | | 2% | 0.329 | 0.276 | 0.366 | 0.085 |

data with diversity criteria is superior. In addition, compared to only selecting high-quality data, the criteria that combine quality and diversity can achieve better performance than simply selecting high-quality data.

**Importance** It is non-trivial to accurately identify and utilize datapoints that significantly affect model performance As shown in Tab. 4, the importance-based data selection approaches often maximize the final performance by integrating the data selection and model optimization processes. To address the challenges of potential huge computation overhead brought by the implementation framework, they propose to use instance-wise data impact and propose efficient parameterization. Moreover, by performing importance resampling in the feature space that depicts geometric structures, they select examples of both high importance and similarity to the target distribution, thereby enhancing the performance on the target tasks. Existing work has confirmed that the importance sampling-based approach can effectively improve the performance on the target tasks with enhanced capabilities of LLMs.

**Hybrid Selection** It is noteworthy that many data selection methods such as LIFT [Xu et al. (2023b)], FL [Bhatt et al. (2024)], DEITA [Liu et al. (2023b)], and QDIT [Bukharin & Zhao (2023)], attempted to combine multiple aspects of data assessment into their selection pipelines. Most of them emphasize the overall assessment of instruction data in that the definitions of "good data" are varied under different scenarios in the era of LLMs. A unitary measurement would inevitably cause a biased selection of datapoints and thereafter leads to the degraded performance of LLMs. Technically, existing hybrid selection methods can be categorized into: 1) parallel setups, and 2) sequential setups. For the former, an operation of maximization or weighted sum is performed on scores or proxy indicators (i.e., losses and gradients) from multiple aspects (e.g., quality and diversity) [Xu et al. (2023b); Bhatt et al. (2024); Bukharin & Zhao (2023)]. For the latter, a sequential setup of quality-, diversity-, or importance-oriented selection techniques is performed step-by-step for hierarchical, stratified filtering [Liu et al. (2023b)]. Comparatively, the parallel setups allow dynamic trade-offs between multiple data aspects by adjusting the aggregation operations. The sequential setups, on the other hand, fail to retrieve the candidates that are filtered out in the preceding quality control steps even if those candidates are of high importance or variety. Therefore, it would be preferred to develop hybrid assessment techniques that simultaneously weigh quality, diversity, and importance. Furthermore, we notice that the importance-based assessment is often overlooked in existing hybrid approaches, implying

that the investigation of integrating importance with quality and diversity is of high potentials for future studies [Zhuang et al. (2024); Lv et al. (2024)].

**Distinctions and Connections**   The methods that are respectively reported in Tabs. 2, 3, and 4 share the similar philosophy of data assessment and selection but differ in the detailed implementations.

For the quality-based selection, most of existing methods like IFD [Li et al. (2023a)], PPL [Ankner et al. (2024)], AutoDS [Zhang et al. (2024c)], and QuRator [Wettig et al. (2024)] mainly use the model-based indicators to measure the quality of each datapoint. However, their indicators are derived from different perspectives such as the perplexity of LLMs and the predicted quality score of a regression model. Both the Alpagasus [Chen et al. (2023b)] and BSDetector [Chen & Mueller (2024)] mainly employ ChatGPT scoring for quality evaluation. Alpagasus directly prompts GPT to score and filter out datapoints, but BSDetector considers both the consistency of the generated responses from GPT and the quality of the ground-truth response. On the other hand, LIFT [Xu et al. (2023b)], InstructionMining [Cao et al. (2023)], and FL [Bhatt et al. (2024)] develop composite methods that involve hand-crafted or model-based indicators, together with GPT scoring, for quality verification of the selected datapoints.

For the diversity-based selection, both DEITA [Liu et al. (2023b)] and ClusterClip [Shao et al. (2024)] adopt the geometry-based measurement where the geometric structure of datapoints is maintained via clustering. DEITA measures the individual-level relationship between one datapoint and its closest neighbor for sampling while ClusterClip simply performs the cluster-level uniform sampling with the quantity limit. QDIT [Bukharin & Zhao (2023)] performs the iterative selection where each step only one most promising datapoint from the remaining dataset is selected to maximize the overall diversity of the selected subset. Despite the fact that DQ [Zhou et al. (2023)] also exploits iterative coreset sampling, it first divides the entire dataset into multiple subset bins and performs sampling across bins for aggregation and deduplication.

For the importance-based selection, DsDm [Engstrom et al. (2024)] and MATES [Yu et al. (2024)] employ model-based indicators to estimate the influence with datamodels. However, these two methods choose different implementations of datamodels. DsDm develops a linear model and approximates the loss of each datapoint via the TARK estimator while MATES directly leverages a small language model to learn to predict the individual influence of each datapoint. DSIR [Xie et al. (2023)], on the contrary, bypasses the need to maintain a specific datamodel. Instead, it estimates the importance of each datapoint by comparing the distributions of the entire selected subset with those of the validation set. Both Skill-it [Chen et al. (2024b)] and LESS [Xia et al. (2024a)] pinpoint the samples that resemble the most to the target set. Skill-it performs sampling via minimizing the evaluation loss while LESS utilizes the gradient similarity as a proxy.

# 7 Future Directions: Challenges and Opportunities

In this section, we present the existing challenges and potential solutions to developing advanced data assessment and selection methods.

## 7.1 Benchmarking Instruction-Tuned LLMs

**There exists a gap between the effectiveness of data selection and the reported performance on benchmarks.**   In existing researches, the ablation studies on the effectiveness of assessment and selection methods are often carried out by comparing the performance of LLMs fine-tuned with the selected and the full dataset. However, for coreset sampling methods that use losses and gradients as proxies for data quality, the downstream performance may not be positively correlated with the selection effectiveness. The reason behind is that the evaluation loss itself [Yang et al. (2022); Hoffmann et al. (2022); Kaplan et al. (2020)] is not informative enough for universal estimation of benchmark performance. [AI@Meta (2024)] demonstrates that the correlation between the negative log-likelihood loss on downstream tasks and the accuracy metrics should be modeled task-by-task and model-by-model. In the light of this statement, it is impractical to simply count on losses or gradients to pinpoint the most beneficial data for improving the downstream performance, let alone methods that try to predict the loss based on various indicators [Cao et al. (2023)]. Furthermore, even if the metrics are exhaustively computed for the selection of each sample,

Table 4: Experimental results of importance-based selection methods are directly cited from their papers.

| Methods | Training Set | Model | Ratio/Size | Reported Results on Testing Set | | | | |
|---|---|---|---|---|---|---|---|---|
| DsDm [Engstrom et al. (2024)] | C4 | Chinchilla-optimal-1.3B | | COPA | OBQA | PIQA | CBT | Hellaswag |
| | | | Random | 0.620 | 0.334 | 0.689 | 0.864 | 0.449 |
| | | | DsDm | 0.630 | 0.312 | 0.690 | 0.882 | 0.423 |
| | | | | Winogrande | BoolQ | COQA | ARC-E | TriviaQA |
| | | | Random | 0.522 | 0.549 | 0.188 | 0.448 | 0.037 |
| | | | DsDm | 0.511 | 0.580 | 0.255 | 0.476 | 0.071 |
| MATES [Yu et al. (2024)] | C4 | Pythia-410M | | SciQ | ARC-E | ARC-C | LogiQA | |
| | | | Random 20% | 0.641 | 0.402 | 0.256 | 0.247 | |
| | | | MATES 20% | 0.660 | 0.418 | 0.250 | 0.257 | |
| | | Pythia-1B | Random 20% | 0.658 | 0.437 | 0.256 | 0.275 | |
| | | | MATES 20% | 0.673 | 0.449 | 0.259 | 0.287 | |
| | | Pythia-410M | | OBQA | BoolQ | HellaSwag | PIQA | Winogrande |
| | | | Random 20% | 0.294 | 0.589 | 0.397 | 0.671 | 0.506 |
| | | | MATES 20% | 0.308 | 0.606 | 0.410 | 0.687 | 0.527 |
| | | Pythia-1B | Random 20% | 0.318 | 0.602 | 0.438 | 0.689 | 0.507 |
| | | | MATES 20% | 0.322 | 0.609 | 0.453 | 0.695 | 0.524 |
| DSIR [Xie et al. (2023)] | The Pile | RoBERTa-Base (125M) | | MNLI | QNLI | QQP | RTE | |
| | | | Random 51.2M | 0.826 | 0.869 | 0.896 | 0.674 | |
| | | | DSIR 51.2M | 0.831 | 0.891 | 0.898 | 0.751 | |
| | | | | SST-2 | MRPC | CoLA | STS-B | |
| | | | Random 51.2M | 0.901 | 0.874 | 0.494 | 0.886 | |
| | | | DSIR 51.2M | 0.905 | 0.877 | 0.540 | 0.892 | |
| Skill-it [Chen et al. (2024b)] | RedPajama | GPT-Neo-3B | | ARC-C | ARC-E | BoolQ | COPA | |
| | | | Skill-it 1B | 0.346 | 0.612 | 0.682 | 0.820 | |
| | | | Uniform 1B | 0.354 | 0.652 | 0.689 | 0.810 | |
| | | | Skill-it 1B | 0.349 | 0.617 | 0.686 | 0.810 | |
| | | | Uniform 1B | 0.353 | 0.624 | 0.677 | 0.800 | |
| | | | Skill-it 1B | 0.348 | 0.620 | 0.687 | 0.810 | |
| | | | Uniform 1B | 0.346 | 0.625 | 0.672 | 0.810 | |
| | | | | HellaSwag | LAMBADA | PIQA | Winogrande | |
| | | | Skill-it 1B | 0.637 | 0.670 | 0.750 | 0.639 | |
| | | | Uniform 1B | 0.639 | 0.644 | 0.748 | 0.628 | |
| | | | Skill-it 1B | 0.639 | 0.667 | 0.752 | 0.632 | |
| | | | Uniform 1B | 0.638 | 0.659 | 0.755 | 0.639 | |
| | | | Skill-it 1B | 0.639 | 0.660 | 0.757 | 0.631 | |
| | | | Uniform 1B | 0.640 | 0.668 | 0.750 | 0.634 | |
| LESS [Xia et al. (2024a)] | Mixed | LLaMA2-7B | | MMLU | TYDIQA | BBH | | |
| | | | Full | 0.516 | 0.540 | 0.432 | | |
| | | | Random 5% | 0.465 | 0.527 | 0.389 | | |
| | | | LESS 5% | 0.502 | 0.562 | 0.415 | | |
| | | LLaMA2-13B | Full | 0.545 | 0.543 | 0.508 | | |
| | | | Random 5% | 0.534 | 0.530 | 0.470 | | |
| | | | LESS 5% | 0.540 | 0.546 | 0.506 | | |
| | | Mistral-7B | Full | 0.604 | 0.577 | 0.530 | | |
| | | | Random 5% | 0.600 | 0.569 | 0.545 | | |
| | | | LESS 5% | 0.618 | 0.603 | 0.560 | | |

the gains brought by one sample might be limited in few tasks. Therefore, to comprehensively reflect the effectiveness of sample selection, the evaluation of instruction-tuned models should be accompanied by the specialised evaluation of the selected datapoints. For the former, all sorts of evaluation strategies have been proposed to precisely evaluate the LLMs [Melis et al. (2017); Chang et al. (2024); Xu et al. (2022); Liang et al. (2022)]. The multiple-choice QA tasks are not enlightening in judging if the instruction-tuned model truly understands the problem rather than simply memorizing the answer choices given the instruction context. For the later, a benchmark for documenting and comparing the statistics of the selected instruction-response pairs in terms of quality, diversity, and importance needs to be constructed in the future. It would benefit the task-wise customized data selection according to the statistical indicators on such a benchmark.

**Test set contamination should be considered during instruction data selection.** For instruction tuning on publicly released pre-trained LLMs, it cannot be too careful to check the potential data leakage where the testing instructions are already modeled during pre-training [Rae et al. (2021); Li et al. (2023b); Magar & Schwartz (2022); Carlini et al. (2019); Marone & Van Durme (2024); Deng et al. (2023); Cao et al. (2024); Jiang et al. (2024b); Magar & Schwartz (2022)]. To improve the performance of pre-trained models on downstream tasks, datapoints (i.e., instruction-like conversations) are already added into the annealing phase of pre-training [AI@Meta (2024); Bilibili (2024); Yang et al. (2024); Bai et al. (2023)]. Therefore, potential risks of data contamination are raised for benchmarking the fine-tuned LLMs. To avoid the negative effect of data leakage on evaluation of the data assessment and selection, it is encouraged to follow [Li et al. (2023a)] to adopt the pre-trained model for experiencing the datapoints before fine-tuning. If the model exhibits overfitting behaviors (i.e., accurately generating the instruction part or producing the same answer choice even with permutation on the choice letters), data contamination is likely to exist and thereafter the testing set should be replaced. For future studies, it would be more reliable to decouple the evaluation of data selection and that of fine-tuned LLMs, where the performance consistency between these two evaluation results can be analyzed to rule out the possibility of contamination.

## 7.2 Unveiling the Definitions of Good Data

**What signifies the most a good datapoint remains an open question.** Unfortunately, there exists no unified criteria on discriminating "good" instructions from "bad" ones. Essentially, the definitions on the general data "quality" differ from task to task and domain to domain [Evans & Murshudov (2013); Flach (2012); Albalak et al. (2024)]. Although existing quality measurement methods can be categorized in terms of quality, diversity, and importance under the present study, they all exhibit more or less ad-hoc properties in methodology. First, studies on instruction tuning are often targeted at improving the performance of LLMs on downstream task. No matter whether these tasks are of general-purpose (e.g., common NLP tasks on leaderboard [Myrzakhan et al. (2024); Wolf et al. (2019)]) or domain-specific applications, such task-orientated data selection itself is only a "proxy" for exploring the underlying "quality" measurement. Especially for coreset sampling methods that directly employ the evaluation set or testing set for distribution matching or importance estimation, instructions that resemble the most to the testing set or bring about performance gains are judged as "good" data. However, such "good" data cannot be easily transferred to another LLM of completely different architecture and parameters. Each time the entire pipeline has to be enforced for a novel task, making it difficult to accumulate universally-acknowledged high-quality data for archiving. Second, each method has an individual quality evaluation system and very few of them ever tried to justify their design and interpret the philosophy behind. It is difficult to validate whether certain component of the selection pipeline can be replaced or removed for better serving a new task-of-interest.

Accordingly, further academic explorations include: 1) to present a more unified, generally applicable definitions on "good" datapoints in terms of fine-grained aspects, and 2) to improve interpretability and explanability of the selection pipeline beyond empirical design.

**The expected model behavior retrospectively determines the trade-off between quality, diversity, and importance for data selection.** The three aspects we used to categorize data assessment methods are actually overlapping with each other, where the "boundary" between two measuring dimension is often hard to explicitly defined. Under such circumstance, the definition of good data can be perceived as the weighted, biased mixture of quality, diversity, and importance. Existing methods are not flexible in dynamically adjusting the mixing weights to adapt to different downstream tasks. Instead, their priority order of the three dimensions is implicitly encoded into the selection of instructions. For instance, [Liu et al. (2023b)] emphasizes quality and importance equally by first establishing the relative ranking of all samples in both quality and complexity. The subset is formed by consecutively selecting the top-ranked samples in sequence, with diversity intervened via ruling out heavily homogenized examples. Such hard-coded, greedy treatment to quality, diversity, and importance is not applicable to scenarios where the behavior of LLMs is expected to cater to varied preference.

In general, the data assessment and selection methods that can adapt to the model requirement under different application scenarios are yet to be systematically developed. For generation tasks like role playing and creative writing, the preferred instruction tuning datapoints should be distinct from those for discriminative tasks like named entity recognition and sentiment analysis.

## 7.3 Scaling Up Datasets

**The optimal scale of the selected subset becomes less explicit with the expansion of datasets.** In the analysis of the disadvantages in exploiting the entire instruction dataset for alignment, putting aside the issue of long training time, we notice that the performance of fine-tuning the entire dataset might not be the optimal. There often exists a critical point of the best selection proportion, and such proportion varies from dataset to dataset. When more instruction datasets from diverse domains and tasks are incorporated, it becomes more difficult to nail down the best selection proportion for three main reasons. First, a large proportion of noise exists in the open datasets and few noisy samples can already cause tremendous negative impact on performance [Song et al. (2022)]. During the pre-processing of instruction dataset, noise can be unintentionally introduced in instruction preparation (e.g., missing context or system prompt), response generation (e.g., unverified or mismatched answers), format wrapping (e.g., invalid `JSON` and unresolved `code`), and text augmentation (e.g., synonym replacement and reorder of words). Second, for specialized tasks in

"vertical" domains, the overfitting of specific prompts occurs [Ma et al. (2023)] when the diversity of input instructions is rather limited. Despite the accuracy and rationality of the instruction-response pairs, LLMs tend to overfit certain patterns of the input instruction rather than truly comprehend the task. Therefore, the increase of dataset size instead reduces the generalization of trained LLMs with lower instruction following capabilities. Third, the forgetting [Zhang & Wu (2024); Jin & Ren (2024b); Wang et al. (2024b)] becomes a severe problem when more instruction datasets are introduced without setting a proper re-playing schedule of pre-training or previously visited instruction tuning datapoints [Parmar et al. (2024); Jin et al. (2021); Ibrahim et al. (2024)]. The skill cultivation of a LLM on any new instruction task heavily relies on its preceding skills acquired during pre-training or previous fine-tuning. Consequently, the expansion of samples for high-level skills would "dilute" those for low-level skills and degrade performance.

To sum up, with the dataset scaling up, fine-tuning with the selected subset instead of the entire set becomes a must-have strategy. To help determine the optimal selection ratio, we suggest the following three guidelines: 1) One may first develop a complex quality measure scheme that uses both indicators and human verification to estimate the noise percentage of each constituting dataset. Without lose of generality, random sampling [Xia et al. (2024b)] can be performed to accelerate quality measurement. To combat noise, a lower ratio (i.e., smaller $S_b$) should be considered for data selection from the noisy $S$. 2) To combat overfitting, both the diversity of datapoints within and across datasets should be emphasized. A higher keeping proportion should be established for datasets with diverse instruction styles, prompt formations, and response patterns, which helps improve the model's instruction following capabilities. 3) For continual fine-tuning, datasets that share similar distributions with pre-training and previous fine-tuning datapoints should be kept to fight against forgetting. The optimal selection ratio and proportion for each dataset is built upon the meticulous and thorough analysis on each instruction dataset, and therefore case-by-case adjustment is needed. For future studies, one may investigate the automation of assessment and selection recipe to minimize the human intervention.

**The optimization of a scalable pipeline for data assessment and selection is of urgent need.** In consideration of the cost of building human-labeled and human-verified instruction tuning datasets, methods that employ powerful LLMs like GPT4 for instruction synthesis [Bradley et al. (2023); Li et al. (2023f); Xu et al. (2023a); Li et al. (2024a); Zhao et al. (2024a); Dong et al. (2024)] have gained increasing attention. The synthetic instructions proliferate cost-effectively with fine-grained control of characteristics such as difficulty and style. Therefore, it is expected to witness a surge of datapoints (e.g., tens or even hundreds millions) in the short future. In that case, datasets of such quantity pose a significant challenge to the robustness, efficiency, and precision of the selection methods. Previous studies like DSIM [Xie et al. (2023)] demonstrated that cheap approximation of features by bag-of-n-grams achieves similar performance but requires much less computing resources. For future research, one may draw inspiration from the data deduplication and filtering techniques in handling billions of pre-training tokens. Especially for the measurement of diversity, the computing of embedding-based pairwise similarity and clustering can be greatly reduced with simplified representations. In addition, the hierarchical philosophy [Hmida et al. (2016); Talavera (1999); Cabezas et al. (2023); Ran et al. (2023)] might be a promising approach to select data from coarse-grained to fine-grained structures. One may apply the devide-and-conquer strategy to recursively handle each subset of the instruction dataset, limiting the peak resource consumption under budget.

## 7.4 Scaling Up LLMs

**The cost-efficiency of data assessment and selection diminishes with larger LLMs involved in the pipeline.** The model-based indicators and coreset sampling methods often require the language model itself to be involved for computation of metrics [Li et al. (2023a)], losses [Chen et al. (2024b)], and gradients [Xia et al. (2024a)]. With the increase of model sizes, it becomes more and more cumbersome to implement the entire pipeline for quality measurement and selection. To expedite the process, one important direction for future study is to develop proper efficient proxy models. Small proxy models have been successfully applied in accelerated fine-tuning of language models [Hoffmann et al. (2022); Liu et al. (2024a)], filtering datasets by perplexity [Ankner et al. (2024)], intervention on retrieval-augmented generation [Tan et al. (2024b)], and performance prediction [Anugraha et al. (2024); Ngu et al. (2024)]. Such proxy models often share the same architecture design with the LLMs under development but own much less parameters. The scaling

law [Kaplan et al. (2020)] confirms the expected consistent behavior between data quantity and model scale, providing practical guidelines on the development of such proxy LLMs.

On the other hand, under the context of data evaluation, it calls upon on rethinking of traditional machine learning techniques such as efficient optimization tricks and dimensionality reduction approaches. For example, in the assessment of loss-based datapoint influence [Feldman & Zhang (2020)], the exhaustive measurement on the marginal performance by moving-each-sample-out and model re-training can be simply approximated by iterative batch-wise sampling tricks with a greedy principle behind. For efficient assessment, PCA [Xu et al. (2023b)] and random projection [Xia et al. (2024a); Park et al. (2023)] are popular choices for obtaining low-rank representations of embeddings and gradients, which facilitates not only metric computation but also storing of datapoints.

**The marginal benefits of instruction tuning diminishes with increasing size of LLMs for knowledge supplement.** Recent studies on the effectiveness of instruction tuning in injecting task-specific or domain-specific knowledge into LLMs [Shi & Lipani (2023); Goyal et al. (2023); Zhang et al. (2024b); Yıldız et al. (2024)] show that the stand-alone instruction tuning might not be the most appropriate method. Compared with strategies like continual pre-training [Cossu et al. (2024); Ke (2024); Cossu et al. (2024)] and instruction modeling [Lou et al. (2024); Cheng et al. (2024); Wang et al. (2022); Xu et al. (2024); Shi et al. (2024)], instruction tuning counts the response sequences for loss computation without sufficient perception of instruction context. For specialized domains like medicine, finance, and laws, if the pre-trained LLMs are in lack of the prerequisite knowledge, the instruction tuning cannot properly activates the parameterized "memory" for alignment but only causes overfitting of the given prompt. In that case, the benefits of data selection are limited with poor generalizability.

Another noteworthy phenomenon in data assessment and selection studies is that due to limited budgets of computing resources, most of the experiments are performed on LLMs of small and moderate size (e.g., less than 7B) to validate the effectiveness of the quality measurement and the selection strategy. Small pre-trained LLMs, by their nature of small parameter size, are more sensitive to the instruction datasets during fine-tuning or continual learning [Schick & Schütze (2020); Yıldız et al. (2024)]. They exhibit the most significant rates of both forggeting (old knowledge) and learning (new knowledge). In the light of such statement, small LLMs tend to sacrifice the task-irrelevant knowledge in return for rapid adaptation towards novel domains and tasks. The selected datasets by various quality measures can impose immediate effect on the parameters of small LLMs, but may weaken on those of huge ones. It remains unknown whether the same quality measurement and data selection pipeline can achieve similar performance gains on both small and large LLMs. For future research of data assessment and selection, extensive experiments are required to validate their efficiency on huge LLMs (e.g., 70B and 405B) [AI@Meta (2024)] and LLMs of mixture-of-experts (MoE) architectures (e.g., Mixtral 8x22B) [Jiang et al. (2024a)].

In consideration of the pre-training corpus, extremely large LLMs already experienced a vast amount of multi-lingual, multi-domain web texts during pre-training, and therefore the priority of the dimensions in data assessment (i.e., quality, diversity, and importance) differs from small LLMs. The association between the model scale and the data selection criteria is yet to be studied.

### 7.5 Validating the Bias and Fairness of Instruction-tuned LLMs

As mentioned in the Sec. 1.3, most existing data selection methods fail to specifically study the dataset bias for consideration of fairness. It is noted that for domain-specific tasks such as solving programming and maths problems, the data bias might not be that severe. However, for general question-answering, recommendation, and creative writing tasks, even ChatGPT could produce biased answers that are intolerable in domains like education [Doan et al. (2024); Zhang et al. (2023b); Chisca et al. (2024); Gao et al. (2024)]. To improve the fairness of LLMs, it is necessary to take the measurement of data bias into serious consideration. Future work includes: 1) the integration of embedding-based (e.g., WEAT [Caliskan et al. (2017)], SEAT [May et al. (2019)]), probability-based (e.g., DisCo [Webster et al. (2020)], CrowS-Pairs Score [Nangia et al. (2020)]), and generated texts-based (e.g., Co-Occurrence Bias Score [Bordia & Bowman (2019)], Full Gen Bias [Smith et al. (2022)]) techniques for fairness evaluation of datapoints [Gallegos et al. (2024)], and 2) the construction of

bias and fairness benchmarks [Jiao et al. (2024)] under various domains and tasks as supplementary kits to the current evaluation pipeline that only focuses on testing the capability of LLMs.

## 8 Conclusion

In this study, we have thoroughly examined the state-of-the-art data assessment and selection methods for instruction tuning of LLMs. The present review presents a unified organization and categorizes these methods in terms of measuring dimensionality: quality, diversity, and importance. In each dimensionality, we outline the representative strategies in details and describe the factors to consider when selecting data for instruction tuning. Furthermore, we report the performance of typical data selection methods and provide discussions on the comparison between these methods. Last but not least, the existing challenges and potential solutions for future studies are summarized in hope for benefitting the research community.

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
