# OpenReview forum: "Unleashing the Power of Data Tsunami: A Comprehensive Survey on Data Assessment and Selection for Instruction Tuning of Language Models"
_TMLR — Accepted by TMLR_

### Review · Reviewer_KHBS · 2024-09-12

**Summary Of Contributions:**

This paper presents a thorough survey on data assessment and selection methods in the instruction tuning of large language models (LLMs). It systematically categorizes the existing approaches into three key dimensions: quality, diversity, and importance. The paper provides a detailed comparison and analysis of these methods, contributing valuable new insights and perspectives on optimizing LLMs performance through effective data selection. Furthermore, it addresses the challenges faced in practical applications and proposes potential directions for future research.

**Audience:**

Yes

**Claims And Evidence:**

Yes

**Requested Changes:**

Please refer to the Weaknesses.

**Strengths And Weaknesses:**

**Strengths:**

1. The paper offers profound insights and a comprehensive analysis of data assessment and selection methods aimed at enhancing LLMs performance. The innovative approach of categorizing existing methods into three dimensions is noteworthy.
2. The inclusion of a detailed link to related content, which categorizes the corresponding papers and provides summaries, significantly enhances the paper's impact within the LLMs community.
3. The survey is both thorough and well-organized, with figures and structure that facilitate easy comprehension for readers.

**Weaknesses:**

1. Beyond the three main categories, there are algorithms that are hybrids of two categories, such as LIST, FL, DEITA, and QBIT. It would be beneficial for the authors to discuss these hybrid algorithms and their corresponding categories in a dedicated section.
2. The paper should clarify the distinctions and relationships between the methods in these three categories.
3. In the "Survey Scope" section, the authors claim that previous studies overlook the measurement of data quality. It is recommended that the authors more explicitly differentiate this survey from previous ones and emphasize the novelty of their work.

---

> ### Author Response · Authors · 2024-11-29
> **Explanations on the hybrid method, distinctions and connections between existing methods, and the survey scope**
>
> **C1: Beyond the three main categories, there are algorithms that are hybrids of two categories, such as LIST, FL, DEITA, and QBIT. It would be beneficial for the authors to discuss these hybrid algorithms and their corresponding categories in a dedicated section.**
>
> R1: Thank you for the constructive comments. First, we modify the manuscript to explain how the existing methods are selected and classified under our taxonomy of data aspects. Please see the newly added section "Criteria on Classification and Selection of Existing Methods" in Sec. 6 (Results and Discussions).
>
> Second, **we agree with the reviewer that there exist many methods that consider two or more aspects in their data selection pipelines**. In this case, we provide an additional section in Sec. 6 for discussions on their hybrid approahes (Hybrid Selection).
>
> Third, we believe that the hybrid technique is of high potentials for future work. Data selection based on a single aspect is easily biased and incomprehensive, which cause damages to the generalization capabilities of LLMs.
>
>
> **C2: The paper should clarify the distinctions and relationships between the methods in these three categories.**
>
> R2: First, we provide the general explanations on the distinctions between the concepts of quality, diversity, and importance in Sec. 1.2 of the introduction section. It is noted that these three concepts could be used interchangeably in previous studies. But in the current manuscript, we try our best to distinguish these concepts with definitions newly provided in the Sec. 1.2 and their general-yet-non-rigorous mathematic formulations newly added at the beginning of Secs. 3, 4, and 5.
>
> Second, we modify the manuscript to provide the criteria on categorization of existing data selection methods on instruction tuning. Please refer to the Sec. 6.
>
> Third, to clarify the distinctions and relationships between these methods, we added one new subsection in the Sec. 6 (Distinctions and Connections). Please see the revised manuscript.
>
> **C3: In the "Survey Scope" section, the authors claim that previous studies overlook the measurement of data quality. It is recommended that the authors more explicitly differentiate this survey from previous ones and emphasize the novelty of their work.**
>
> R3: To differentiate the present study with previous surveys, we provide more explanations in the Sec. 1.1 of  the introduction section. Please refer to our revised manuscript for details.

---

> > ### Comment · Reviewer_KHBS · 2024-12-08
> >
> > Thanks for your responses. I have carefully reviewed your responses and the revised manuscript. Most of my concerns have been adequately addressed.

---

### Review · Reviewer_BZqh · 2024-09-24

**Summary Of Contributions:**

The survey examined data assessment and selection method for instruction tuning of LLMs. It presents a taxonomy based on three aspects: quality, diversity, importance. Within each aspect, further categorization of method is given and representative method are summarized and discussed. The performance of some representative method are summarized and discussed per aspect. Finally, the survey points out issues of existing methods and provide future directions.

**Audience:**

Yes

**Broader Impact Concerns:**

LLMs are known to have bias/fairness issues, author may consider adding discussion on how these are handled by data assessment and selection work currently.

**Claims And Evidence:**

Yes

**Requested Changes:**

1. Related surveys are collected in the survey, but a compare and contrast is lacking.
2. Figure 1 is not vector graph, small texts are hard to read.
3. The mathematical formulation in Preliminaries Instruction Dataset Preparation is not clear to me.
    1. instruction and input are used interchangeably while considered as two parts in first sentence. Also, what is x? is x the same as I? and what is the relationship between p and x?
    2. The equation 2 is also confusing, the input of sampling mechanism \pi takes input b and a single data point that is a result of argmax. Author should correct the formulation. Also \S_{b}^{\*} ‘s meaning is unclear.
The terminology is not unified across the survey. datapoint and instruction datapoint are also used interchangeably.
4. Eq (6):  the indicators should not be sharing parameters in general?
5. Eq (8): I don’t understand why small \tilde{S}(x_i) here corresponds to “easy”, it should be the opposite since the perplexity would need to be small, no? (correct me if I am wrong)
6. The sentence below equation 10 “where yi(j) ∈ R^{N_{vocab}} denotes the one-hot vector of the vocabulary size N_vocab,” . The wording here is confusing.
7. Quality-based, diversity-based and importance-based method are discussed, but author did not elaborate the definition of those quantities. Also I think some high level rationale should be provided regarding what these three aspects are and why use them to build the taxonomy. I understand these concepts overlap practically, but some discussion of the usage of these concepts within the scope of the survey is appreciated.
    A general formulation of methods in each of these aspects would be beneficial if possible. This might help in resolve the possible confusion of the concept for example in Ch 4, diversity seems to refer to lexical and semantic richness, but why bias, fairness aspects of diversity are not discussed in the survey is not clear.

8. Data selection and assessment targeting bias, fairness finetuning is completely missing in this survey, I wonder the authors opinion on these.

9. I suggest author to recheck their literature summary across the draft, some summaries are wordy but fail to convey main idea of surveyed work. Take an example of Jiang et al. (2023b) referred in 5.2 the paragraph below equation 42, I have a hard time understanding the calibration process despite it taking a whole paragraph.

10.  How are table 1, 2, 3 constructed? How do authors pick methods to be summarized to the tables?

11. Again, the draft should be checked thoroughly by the authors, as the current version contains a lot of typos, unfinished sentences, or empty references.

**Strengths And Weaknesses:**

Strength:
1. The survey provide good collection of LLM data selection method that focus on three aspects: quality, diversity and importance. The survey nicely summarized representative methods of hand-crafted and model-based methods that is easy to understand and refer to for LLM community.

2. The provided suggestions on future directions are insightful.

Weakness:
1. This draft contains a lot of typos and unfinished sentences.
2. see requested changes.

---

> ### Author Response · Authors · 2024-11-29
> **Explanations on mathmatical formulations, survey scope, related work, and discussions**
>
> **C1: This draft contains a lot of typos and unfinished sentences.**
>
> R1: We have carefully corrected those typos and polished the English sentences.
>
>
> **C2: see requested changes.**
>
> R2: We have made changes according to the comments and suggestions.
>
>
> **C3: Related surveys are collected in the survey, but a compare and contrast is lacking.**
>
> R3: Thank you for the suggestions. We have revised the manuscript to describe the differences between our survey and the related surveys. Please see the revised Sec. 1.1 (highlighted in blue) for our explanations on the comparison and contrast.
>
>
> **C4: Figure 1 is not vector graph, small texts are hard to read.**
>
> R4: Thank you for the comment. We have adjusted the Figure 1 to make texts easy to read.
>
>
> **C5: The mathematical formulation in Preliminaries Instruction Dataset Preparation is not clear to me.**
>
> R5: For the first question, the $x$ is not the same with $I$. To clarify it, we would like to present an example for explanations.
>
> The $I_i$ denotes the raw text of the $i$-th sample in the dataset. It usually contains three components: 1) instruction, 2) input, and 3) response. Take one example from the popular Alpaca dataset:
> ```
> {
>     "instruction":"Explain the difference between a URL and a domain name."
>     "input":""
>     "output":"A URL, which stands for Uniform Resource Locator, is the entire address used to visit a website or access a specific online resource. It includes the protocol (such as HTTP or HTTPS), the domain name, and other elements like the path, parameters, or anchor. For example, the URL of this page is https://play.aidungeon.io/main/homePage.\n\nOn the other hand, a domain name is a component of a URL that specifically identifies the website or web page. It is the part of the URL that comes after the protocol and before the path. For example, in the URL above, the domain name is \"play.aidungeon.io\". \n\nIn simpler terms, you can think of a URL as an address that leads you to a specific location on the web, while a domain name is the specific name of the location you're visiting."
> }
> ```
>
> The $p_i$ is the prompt (in the text form) for wrapping the above text into instruction tuning pairs of one input and one response. The structure of wrapping, namely the chat template, depends on the model families under development (e.g., LLaMAs, Mistrals, and Qwens). Let's take the Qwen model families as an example. Its wrapping method is defined in the  of "tokenizer_config.json" below.
>
> ```
> {
>     "chat_template": "{% for message in messages %}{% if loop.first and messages[0]['role'] != 'system' %}{{ '<|im_start|>system\nYou are a helpful assistant.<|im_end|>\n' }}{% endif %}{{'<|im_start|>' + message['role'] + '\n' + message['content'] + '<|im_end|>' + '\n'}}{% endfor %}{% if add_generation_prompt %}{{ '<|im_start|>assistant\n' }}{% endif %}"
> }
> ```
>
> Since the instruction and the input parts in $I_i$ are from the role of users and the output part in $I_i$ is from the role of assistant, we can generate the wrapped prompt $p_i$ as the following:
> ```
> <|im_start|>system\nYou are a helpful assistant.<|im_end|>\n<|im_start|>user\nExplain the difference between a URL and a domain name.<|im_end|>\n<|im_start|>assistant\nA URL, which stands for Uniform Resource Locator, is the entire address used to visit a website or access a specific online resource. It includes the protocol (such as HTTP or HTTPS), the domain name, and other elements like the path, parameters, or anchor. For example, the URL of this page is https://play.aidungeon.io/main/homePage.\n\nOn the other hand, a domain name is a component of a URL that specifically identifies the website or web page. It is the part of the URL that comes after the protocol and before the path. For example, in the URL above, the domain name is \"play.aidungeon.io\". \n\nIn simpler terms, you can think of a URL as an address that leads you to a specific location on the web, while a domain name is the specific name of the location you're visiting.<|im_end|>
> ```
>
> It is noted that the instruction part (not involved in loss computation) starts with "<|im_start|>system" and ends with "<|im_start|>assistant\n". The response part (involved in loss computation) starts with "A URL, which stands for" and ends with "visiting.<|im_end|>".

---

> > ### Author Response · Authors · 2024-11-29
> > **Explanations on mathmatical formulations, survey scope, related work, and discussions (part2)**
> >
> > R5: --continued--
> > The $x_i$ is the tokenized version of $p_i$ in the form of token_ids. It is noted that $x_i$ is a sequence of numbers with each element indicating the token index of the vocabulary for a given model. The tokenized $x_i$ is presented as follows:
> > ```
> > 151644 8948 1699 2610 525 264 10950 17847 13 151645 1699 151644 872 1699 840 20772 279 6672 1948 264 5548 323 264 7947 829 13 151645 1699 151644 77091 1699 32 5548 11 892 13352 369 47889 11765 98653 11 374 279 4453 2621 1483 311 3947 264 3910 476 2615 264 3151 2860 5101 13 1084 5646 279 11507 320 20805 438 10130 476 61044 701 279 7947 829 11 323 1008 5424 1075 279 1815 11 5029 11 476 17105 13 1752 3110 11 279 5548 315 419 2150 374 3703 1110 1363 13 3779 23452 4245 15351 18215 2665 7110 77 1699 1925 279 1008 1424 11 264 7947 829 374 264 3692 315 264 5548 429 11689 35511 279 3910 476 3482 2150 13 1084 374 279 949 315 279 5548 429 4041 1283 279 11507 323 1573 279 1815 13 1752 3110 11 304 279 5548 3403 11 279 7947 829 374 7245 1363 13 3779 23452 4245 2105 13 1124 77 1699 641 34288 3793 11 498 646 1744 315 264 5548 438 458 2621 429 11508 498 311 264 3151 3728 389 279 3482 11 1393 264 7947 829 374 279 3151 829 315 279 3728 498 2299 16721 13 151645
> > ```
> >
> > It is noted that after tokenization, the instruction part starts from "151644,8948" and ends with "151644,77091,198". The response part starts with "32,5548,11,892,13352,369" and ends with "2682,5853,13,151645".
> >
> > We use the $t$ to denotes the index where the response starts. In this case, we have $x_{i(t)}$=32, the instruction part $x_{i(<t)}$=[151644, 8948, ..., 151644, 77091, 198] and the response part $x_{i(\ge t)}$=[32,5548,11,892,13352,369,...,2682,5853,13,151645].
> >
> > For clearer explanations, we revise the preliminaries and provide an exmaple for clarification.
> >
> >
> > For the second question, we would like to apologize for the mis-understanding caused by Eq. 2. The goal of the dataset selection is to choose a subset $S_b$ from the entire dataset $S$ with $S_b\subset S$. The $b$ denotes the budget for selection, where the size of the subset $S_b$ should be less or equal than $b$.
> >
> > The Eq. 2 represents the process that we perform the sampling operation for at most $b$ times. And at each time, we only sample one of the most promising instruction datapoint $x_i$ according to: 1) the quantitative score $q(x_i)$ for measuring $x_i$, and 2) the sampling algorithm (e.g., greedy or heuristic sampling).
> >
> > We use the $S_{b}^*$  to emphasize that the subset  $S_b$ is selected by sampling based on $q(\cdot)$ and $b$ as an optimization output. To avoid mis-understanding, we discard the $*$ notation and uses $S_{b}$ consistently.
> >
> > To make the formulation easy to understand, we polish the Eq. 2 and provide more explanations.
> >
> > To unify the terminology, we use the $x_i$ to refer to the datapoint. In addition, we use the $x_{i(<t)}$ to refer to the instruction part of the datapoint, and the $x_{i(\ge t)}$ to refer to the response part of the datapoint. We will refrain from the usage of "instruction datapoint" to avoid mis-understanding.
> >
> > **C6: Eq (6):  the indicators should not be sharing parameters in general?**
> >
> > R6: Thank you for the constructive comment. We have modified the formula to allow different parameter representations.
> >
> > **C7: Eq (8): I don’t understand why small \tilde{S}(x_i) here corresponds to “easy”, it should be the opposite since the perplexity would need to be small, no? (correct me if I am wrong)**
> >
> > R7: We appologize for the inappropriate descriptions. We remove the reciprocal (exponential of -1) to make it clearer that datapoints with small $\tilde{S}(x_i)$ are easy ones. Besides, we polish our explanations in the manuscript to indicate the selection principal under different data regimes.
> >
> >
> > **C8: The sentence below equation 10 “where yi(j) ∈ R^{N_{vocab}} denotes the one-hot vector of the vocabulary size N_vocab,” . The wording here is confusing.**
> >
> > R8: We have modified the explanations to make it easy to understand. Please refer to the modified manuscript.

---

> > > ### Author Response · Authors · 2024-11-29
> > > **Explanations on mathmatical formulations, survey scope, related work, and discussions (part3)**
> > >
> > > **C9: Quality-based, diversity-based and importance-based method are discussed, but author did not elaborate the definition of those quantities. Also I think some high level rationale should be provided regarding what these three aspects are and why use them to build the taxonomy. I understand these concepts overlap practically, but some discussion of the usage of these concepts within the scope of the survey is appreciated. A general formulation of methods in each of these aspects would be beneficial if possible. This might help in resolve the possible confusion of the concept for example in Ch 4, diversity seems to refer to lexical and semantic richness, but why bias, fairness aspects of diversity are not discussed in the survey is not clear.**
> > >
> > > R9: Thank you for the suggestions. For the first question, we present the definitions of the three aspects of data, namely quality, diversity, and importance,  in the context of data assessment for instruction tuning of LLMs. Please see our revised introduction.
> > >
> > > For the second question, we would like to explain that it is difficult to cover all the presented methods and their variants using only one mathematic formulation in each aspect. However, from a high level of perception, it is possible to deliver a unified mathematic reformulation for measurement of these aspects in a non-rigorous manner. Please see our revised sections 3,4,and 5 of the manuscript.
> > >
> > > For the third question, we acknowledge that the aspects of bias and fairness are not included into the diversity measurement of the present study. We do not introduce the bias or fairness into the section of diversity-based methods for the following reasons:
> > >
> > > 1) Under the context of data selection for instruction tuning, most existing methods do not even notice the concept of bias or fairness. Therefore, they are in lack of pre-processing and filtering steps that are specifically designed for reducing potential negative impacts of biased training data.
> > >
> > > 2) Existing methods that target at reducing social bias and unfairness of pre-training and fine-tuning datasets [1][2][3] follow their own definitions of diversity in terms of gender, race, religion, profession, age, and political ideology. It is difficult to bring in all these evaluation techniques (e.g., embedding-based, probability-based, and generated texts-based) under our taxonomy of selection methods.
> > >
> > > 3) The concept of bias and fairness overlaps with both quality and diversity control. It is not exclusive to the "diversity" but also implicitly reflected in "quality". For example, to control the harmful contents like hatred and racism responses, existing quality-based selection can use either model-based indicators (e.g., reward model) or GPT scoring for filtering. The publicly released reward models and instruction-tuned LLMs are already aligned with human preference, which adhere to the 3H principles (Helpfulness, Harmlessness, and Honesty). Therefore, datapoints with prejudiced and stereotyped responses will receive lower quality scores and filtered out after quality-based selection.
> > >
> > > 4) The validation of de-bias techniques might not be easily revealed from existing mainstream benchmarks like ARC, HellaSwag, MMLU, AlpacaEval. The evaluation of bias or fairness of the instruction-tuned LLMs is not covered in existing studies, which makes it difficult to judge whether the  selection methods improve the fairness of the tuned LLMs.
> > >
> > >
> > > However, **we agree with the reviewer that bias and fairness should be emphasized in building responsible LLM systems**. There exists a preliminary study [4] that shows instruction-tuned LLMs tend to exhibit more bias than the pre-trained LLMs. Therefore, we would like to spare one subsection in the introduction section and one subsection in the future directions section to respectively: 1) explain the reasons why the bias and fairness aspects are not detailed in the present study, and 2) highlight the importance of bias and fairness for development of comprehensive data selection methods.
> > >
> > > Ref[1] Gallegos, Isabel O., et al. "Bias and fairness in large language models: A survey." Computational Linguistics (2024): 1-79.
> > >
> > > Ref[2] Chu, Zhibo, Zichong Wang, and Wenbin Zhang. "Fairness in large language models: A taxonomic survey." ACM SIGKDD explorations newsletter 26.1 (2024): 34-48.
> > >
> > > Ref[3] Li, Yingji, et al. "A survey on fairness in large language models." arXiv preprint arXiv:2308.10149 (2023).
> > >
> > > Ref[4] Itzhak, Itay, et al. "Instructed to Bias: Instruction-Tuned Language Models Exhibit Emergent Cognitive Bias." Transactions of the Association for Computational Linguistics 12 (2024): 771-785.

---

> > > > ### Author Response · Authors · 2024-11-29
> > > > **Explanations on mathmatical formulations, survey scope, related work, and discussions (part4)**
> > > >
> > > > **C10: Data selection and assessment targeting bias, fairness finetuning is completely missing in this survey, I wonder the authors opinion on these.**
> > > >
> > > > R10: Thank you for raising the constructive question. We have explained the reasons why the bias and fairness aspects are not detailed in the present study. Please see our response to the 7-th comment (above). Accordingly, we have modified our manuscript to make it clearer. Please see the introduction section (Sec. 1.3) and the future directions section (Sec. 7.5).
> > > >
> > > > **C11: I suggest author to recheck their literature summary across the draft, some summaries are wordy but fail to convey main idea of surveyed work. Take an example of Jiang et al. (2023b) referred in 5.2 the paragraph below equation 42, I have a hard time understanding the calibration process despite it taking a whole paragraph.**
> > > >
> > > > R11: We polish the summaries of Jiang et al. (2023b) so that they are easy to understand. We also check the other summaries for preciseness of the surveyed work.
> > > > Specifically for the Jiang et al. (2023b), we remove the detailed explanations on the preparation of answer choices for open-ended generation tasks. Instead, we focus more on the calibration process itself. The new summaries are presented below:
> > > > ```
> > > > \cite{jiang2023calibrating} target at the over-confidence problem of LLMs after instruction tuning~\cite{kadavath2022language}, and propose the CAPE to calibrate the uncertainty with augmented prompt ensembles. They first transform all discriminative and generative tasks into multiple-choice problems, and use the LLM's predicted probabilities over answer choices (e.g., A, B, C) for uncertainty estimation. Then, prompt augmentations are performed via paraphrasing of the template, permutation of the in-context examples and the answer choices. Multiple predictions over the answer choices are collected with the augmented prompts as inputs, which helps calibrate the uncertainty in an ensemble manner. Such calibrated uncertainty tells if an instruction-tuned LLM simply memorizes the response to a given prompt rather than truly understanding the instruction. Therefore, it can be used to precisely choose important datapoints with high uncertainty.
> > > > ```
> > > >
> > > > **C12: 10.How are table 1, 2, 3 constructed? How do authors pick methods to be summarized to the tables?**
> > > >
> > > > R12: First, for the construction of Tables 1, 2, and 3, we directly cite the statistics information such as the used datasets, the model families under investigation, and the reported results on the public benchmarks.
> > > >
> > > > Second, for the classification of different methods in terms of quality, diversity, and importance, we consider the following two aspects: 1) the data characteristics emphasized in the development of their method; 2) the optimization objectives organized in this paper that are closest to those proposed in their papers. For example, DEITA specifically sets a similarity-based filtering step to improve the diversity of the chosen subset. Therefore, it falls under the category of diversity-based selection methods. DSIR, on the other hand, directly estimates the individual importance of datapoints by matching the distribution between the selected subset and the target evaluation set. However, we admit that many of these methods actually design compound, multi-facet algorithms that more or less consider quality, diversity, and importance via sequential steps or balancing weights. In this case, we choose the most representative methods respectively under our categories without misinterpreting their mechanisms.
> > > >
> > > > Third, we select the most recent instruction tuning methods that specifically mention the development of data selection techniques. We believe these representative methods could provide valuable insights into the measurement of data quality, diversity, and importance for reference. In addition, most of the selected methods utilize open-sourced LLMs such as Mistral and LLaMA families for experiments, which would benefit practical applications in consideration of the privacy issues and inference costs brought by the access to close-sourced proprietary models.
> > > > To make the representations of tables easier to understand, we provide explanations in the modified manuscript. Please see the revised section for presenting the results of recent data selection methods.
> > > >
> > > > **C13: Again, the draft should be checked thoroughly by the authors, as the current version contains a lot of typos, unfinished sentences, or empty references.**
> > > >
> > > > R13: We have polished our draft manuscript to make it easier to read. The typos, unfinished sentences, and imcomplete references have been carefully checked for higher readability.

---

### Review · Reviewer_r62g · 2024-11-24

**Summary Of Contributions:**

In this paper, the authors present a survey on data assessment and selection methods for instruction tuning of LLMs. This is an important topic with a possibility of significant impact on future research. The authors systematically categorize the literature into three main perspectives: quality, diversity, and importance. Next, for each category, representative methods from both model-based and hand-based indicators are elaborated. The survey also includes an empirical comparison between different approaches. After introducing related works, the paper proposed a number of open questions for the research community to consider.

**Audience:**

Yes

**Broader Impact Concerns:**

There are no concerns about the broader impact of the work.

**Claims And Evidence:**

Yes

**Requested Changes:**

1. Fix the formatting inconsistencies with quotation marks throughout the paper.
2. I request the authors to provide more detailed descriptions of the algorithms for easier understanding. Although it is optional.

**Strengths And Weaknesses:**

Strengths:

1. The paper does a good and detailed survey of an important topic. Each approach has been explained with clarity.
2. The empirical results and discussions in section 6 covering different approaches, LLMs, and test datasets can be really helpful to the community.
3. Summarized and proposed a list of open research questions for future research.

Weaknesses:

I did not find any major concern with this survey.

1. There are some formatting issues with quotation marks throughout the paper.
2. Some algorithms could benefit from bit more detailed descriptions for easy understanding by a reader without any background in this field.

In general, I appreciate the author's work. Overall, this is a very comprehensive survey analyzing an important field.

---

> ### Author Response · Authors · 2024-11-29
> **Formatting issues and explanations on algorithms**
>
> **C1: There are some formatting issues with quotation marks throughout the paper.**
>
> R1: Thank you for the comment. We have carefully revised the paper to improve the formatting and English writting.
>
>
> **C2: Some algorithms could benefit from bit more detailed descriptions for easy understanding by a reader without any background in this field.**
>
> R2: First, we polish the introduction section to explain the concepts of quality, diversity, and importance mentioned in the present study. We hope that such explanations could help readers get familiar with the background contexts.
>
> Second, we try our best to provide general-yet-non-rigorous formulations of quality-based, diversity-based, and importance-based data assessment at the beginning of Secs. 3, 4, and 5. Althought such formulations may not be strictly precise to embrace all existing methods and their variants, they could inform readers what kinds of objectives are to be achieved respectively in different data aspects.
>
> Third, we provide more detailed explanations on the methods mentioned in Algs. 1~7. We hope that the added explanations could better help readers understand the principles of these methods.
>
>
> **C3: In general, I appreciate the author's work. Overall, this is a very comprehensive survey analyzing an important field.**
>
> R3: Thank you very much.
>
>
> **C4: Fix the formatting inconsistencies with quotation marks throughout the paper.**
>
> R4: We have replaced all the quotation marks with "" to fix the formatting inconsistencies.
>
>
> **C5: I request the authors to provide more detailed descriptions of the algorithms for easier understanding. Although it is optional.**
>
> R5: Thank your for the suggestion. We have provided explanations to make algorithms easier to understand. Please see our revised manuscript.

---

### Decision · Action_Editor_Bew5 · 2024-12-28

**Recommendation:** Accept as is

**Comment:**

Reviewers unanimously agreed to accept, and agreed that the paper was well-written and easy to understand, with nice summaries of previous works. No reviewer found any

There were some notational issues pointed out by Reviewer BZqh's careful reading, but these seem to have been resolved. Reviewer KHBS asked for the paper to clarify its novelty over previous works, and also cover hybrid methods - this was also resolved. Otherwise, no major issues.

There was a 2/3 vote on the survey certification as well - this is a high quality survey, and I agree with the decision for the certification.

**Audience:**

Yes, for LLM practitioners applying instruction tuning.

**Claims And Evidence:**

This is a survey paper, with the following contributions:
  * Listing out multiple different data selection methods for instruction tuning, organized by "quality, diversity, importance"
  * Discussion of pros / cons between different methods, and lists of reported metrics
  * Future research and conclusions

For someone who doesn't research instruction tuning, I found the paper to be well-written and organized, and I learned many new techniques. The listing of methods in every selection method (e.g. quality) was especiall clean - i.e. each cited method was given a brief summary, along with concise and precise equations to distinguish them between others.

Section 1.3 points out also the scope of the paper, i.e. what it doesn't claim to discuss (bias/fairness).

(Small nits) A few typos I found:
* Equation 22 can just be one line.
* Equation 48 to 51's gigantic spacing can be fixed.